# Compact light field photography towards versatile three-dimensional vision

Xiaohua Feng [1,3✉], Yayao Ma[2,3] & Liang Gao [2✉]

Inspired by natural living systems, modern cameras can attain three-dimensional vision via multi-view geometry like compound eyes in flies, or time-of-flight sensing like echolocation in bats. However, high-speed, accurate three-dimensional sensing capable of scaling over an extensive distance range and coping well with severe occlusions remains challenging. Here, we report compact light field photography for acquiring large-scale light fields with simple optics and a small number of sensors in arbitrary formats ranging from two-dimensional area to single-point detectors, culminating in a dense multi-view measurement with orders of magnitude lower dataload. We demonstrated compact light field photography for efficient multi-view acquisition of time-of-flight signals to enable snapshot three-dimensional imaging with an extended depth range and through severe scene occlusions. Moreover, we show how compact light field photography can exploit curved and disconnected surfaces for real-time non-line-of-sight 3D vision. Compact light field photography will broadly benefit high-speed 3D imaging and open up new avenues in various disciplines.

[1] Research Center for Humanoid Sensing, Zhejiang Laboratory, Hangzhou, China. [2] Department of Bioengineering, University of California, Los Angeles, USA. [3] These authors contributed equally: Xiaohua Feng, Yayao Ma. ✉email: fengxiaohua@zhejianglab.com; gaol@ucla.edu

Three-dimensional (3D) imaging is vital for sensing, modeling, and understanding the physical world[1,2], with a broad range of applications in navigation, robotics, and medical imaging[3,4]. However, there is an inherent dimensionality gap between a 3D scene and the recording sensors, which can at most be arranged on a two-dimensional (2D) surface as in curved sensor arrays[5]. As a result, only a 2D projection of the scene can be captured from a given perspective. To recover depths, one must perform additional measurements along an extra axis of light: an angular axis in multi-view measurements or a temporal axis in time-of-flight sensing. While multi-view methods, including stereo[6], structured-light[7], and light field cameras[8], can attain exceptional depth accuracy (<100 μm) at near distances and operate at a relatively high speed, their accuracies degrade quadratically with distance and ultimately fail at a long range[6]. Except for structured light that employs active illuminations, multi-view methods also rely heavily on object texture for effective depth extraction. On the other hand, time-of-flight techniques are agonistic of textures and can maintain the depth resolution over a large detection range[9–11]. However, high-speed and dense depth mapping robust against stochastic motions is still challenging for time-of-flight cameras. The distinct strengths and limitations of multi-view methods and time-of-flight techniques have long divided the design of 3D imaging cameras, confining the capability and application scopes of incumbent 3D vision solutions.

Performing multi-view time-of-flight measurements has potentially disruptive benefits. Besides bringing 3D imaging to an ultrafast time scale[12] and fueling the development of new 3D vision capabilities[13,14], a dense multi-view measurement can substantially bolster the sensing range of a time-of-flight camera[15] and make it possible to see through occlusions[15,16], a recurring challenge for visual tracking in computer vision[17]. However, current methods for acquiring multi-view time-of-flight signals suffer from either a limited number of views[13] or being too time-consuming that hinders dynamic imaging[15]. More problematically, such direct multi-view acquisition of time-of-flight signals exacerbates the "big data" issue. Even with a low spatial resolution, sensing along the extra angular or temporal dimension yields a large amount of data—multi-view measurements generate a plethora of images from different views, and a single time-of-flight (temporal) trace involves thousands of time points. Adding yet another dimension will increase the system complexity and data load so fast that makes real-time image processing and streaming impracticle[16]. Driven by the growing need of high-resolution 3D imaging with large-format detectors like the megapixel SPAD sensors[18], designing efficient multi-view measurements becomes increasingly more relevant. Moreover, sensors for infrared wavelengths, sub-picosecond measurements[19], and other specialized applications[20] are still limited in element counts, which prevent dense 2D image sampling[21] and consequently hamper 3D imaging via conventional multi-view methods.

To address these challenges, we present compact light field photography (CLIP) to sample dense light fields[8] with a significantly improved efficiency and flexibility. Unlike previous compressive light field cameras[22–24] that require densely sampled 2D images for recovering a 4D light field, CLIP is a systematic framework to design and transform any imaging model that employs nonlocal data acquisition into a highly efficient light field-imaging approach: by distributing the designed or existing nonlocal image acquisition process into different views and modeling the correlations inherent in 4D light fields, CLIP can recover the 4D light field or directly retrieve refocused images from a measurement dataset even smaller than a single sub-aperture image. Under the CLIP framework, sensors of arbitrary formats—a single pixel, a linear array, or a sparse 2D area detector

—can be employed for efficient light field imaging by transforming the imaging models of a single pixel camera, $x$-ray computed tomography and a diffuser camera[25], to name a few. Additionally, CLIP is natively applicable to camera array systems, and promotes robustness against defective sensor measurements and severe scene occlusions. With CLIP, we seamlessly synergized multi-view with time-of-flight techniques, and demonstrated single-shot 3D imaging of texture-less scenes in an extended depth range, robust 3D vision through severe occlusions, and real-time non-line-of-sight (NLOS) imaging with curved and disconnected walls, a critical task for field applications not yet fufilled[3,26–31].

## Results

**Principle of compact light field photography.** In linear systems, the image acquisition process can be written in a general matrix formalism as

$$\mathbf{f} = \mathbf{A}\mathbf{h} + \boldsymbol{\sigma},\tag{1}$$

where $\boldsymbol{\sigma}$ is the measurement noise, $\mathbf{h}$ and $\mathbf{f}$ are the vectorized image and measurement, respectively. $\mathbf{A}$ is the $m \times N^2$ system matrix (for an image resolution of $N \times N$ throughout the manuscript), which is square ($m = N^2$) for full-rate sampling and rectangular ($m < N^2$) under compressive sensing. This formulation, though general, typically assumes all $m$ measurements of the scene are obtained from a single view and hence possessing no light field capability. To record light fields with an angular resolution of $l$, the measurement procedure must be repeated either parallelly (via a lens array[32]) or sequentially at $l$ different views[33], leading to a measurement dataset of size $m \times l$. In CLIP, we break this convention by employing nonlocal acquisition for the row entries of matrix $\mathbf{A}$ and splitting the $m$ measurements into $l$ different views, thereby compactly recording light field data with an angular resolution of $l$. As derived in Supplementary Note 5, this transforms the image model into:

$$\mathbf{f} = \begin{bmatrix} \mathbf{A_1} & \cdots & \mathbf{0} \\ \vdots & \mathbf{A_2} & \vdots \\ \mathbf{0} & \ddots & \mathbf{0} \\ \mathbf{0} & \cdots & \mathbf{A_l} \end{bmatrix} \begin{bmatrix} \mathbf{P_1} \\ \mathbf{P_2} \\ \vdots \\ \mathbf{P_l} \end{bmatrix} + \boldsymbol{\sigma} = \mathbf{A}' \begin{bmatrix} \mathbf{P_1} \\ \mathbf{P_2} \\ \vdots \\ \mathbf{P_l} \end{bmatrix} + \boldsymbol{\sigma} = \mathbf{A}'\mathbf{P} + \boldsymbol{\sigma},\tag{2}$$

where $\mathbf{A_k}$ is the $k$th sub-matrix such that $\mathbf{A} = [\mathbf{A_1}; \mathbf{A_2}; \cdots, \mathbf{A_l}]$, $\mathbf{A}'$ is the transformed block-diagonal matrix and $\mathbf{P} = [\mathbf{P_1}; \mathbf{P_2}; \cdots, \mathbf{P_l}]$ is the 4D light field. While one can exploit the sparisity prior to compressively recover a 4D light field at this stage, CLIP can further retrieve a refocused image directly by explicitly modeling the correlations in the 4D light field to better cope with complex scenes (see Supplementary Note 8 for comparisons).

This is inspired by the observation that images of the same scene acquired from different views share the same content in photographic applications (Supplementary Note 1): there is only a depth-dependent disparity between any two sub-aperture images, as illustrated in Fig. 1a. Therefore, one can explicitly model the correlations among the sub-aperture images by digitally propagating the light field, which relates the sub-aperture image at view $k$ (denoted as $\mathbf{P}_k$) to a reference sub-aperture image $\mathbf{h}$ via an invertible shearing operator $\mathbf{B}_k$ as $\mathbf{P}_k = \mathbf{B}_k\mathbf{h}$ (Supplementary Note 1) and the $m$ measurement data

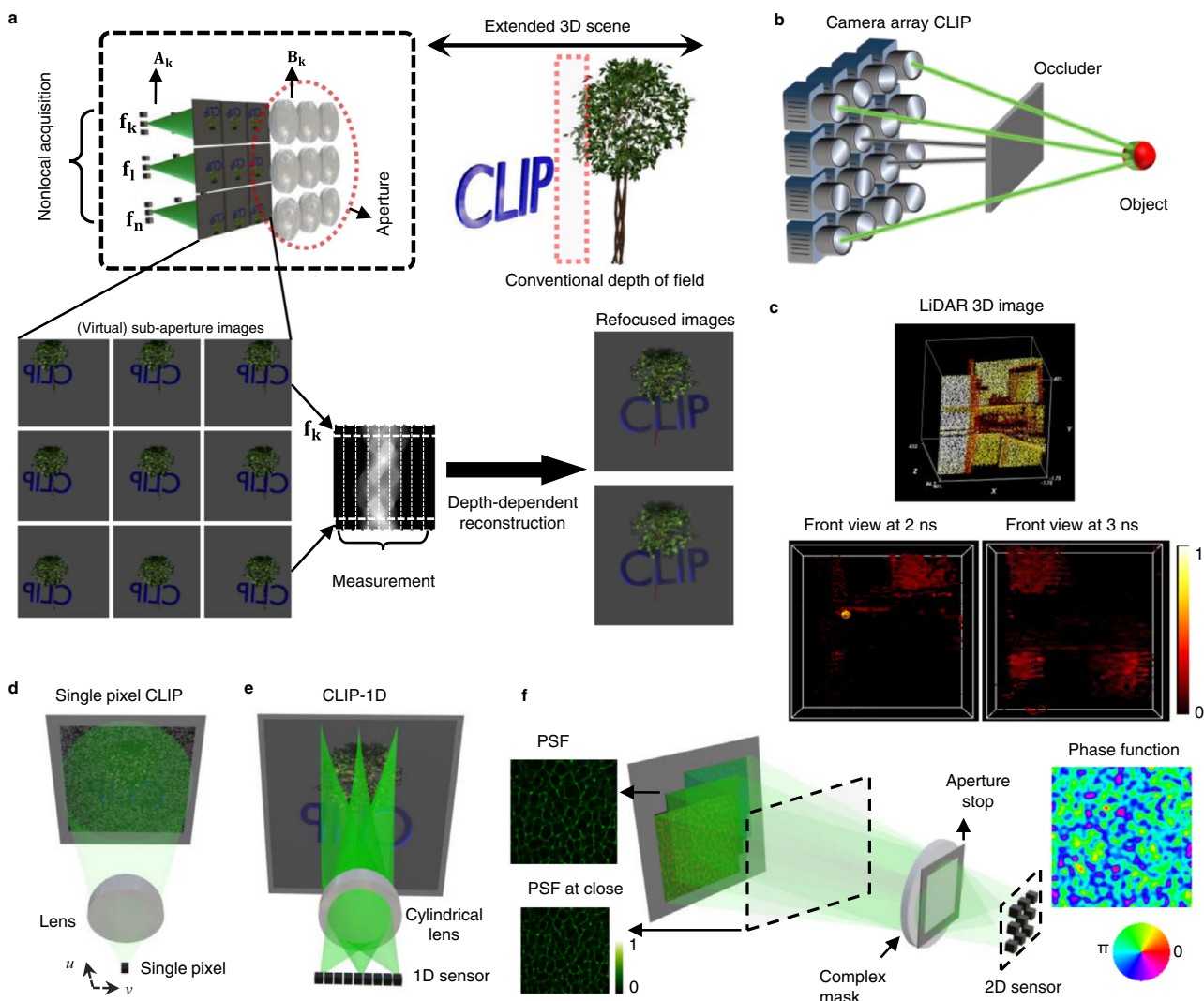

**Fig. 1 Principle of compact light field photography. a** A conventional light field camera captures the scene from different views with a lens array and records all sub-aperture images. In contrast, CLIP records (operator $\mathbf{A}_k$) only a few nonlocal measurements ($\mathbf{f}_k$ to $\mathbf{f}_n$) from each sub-aperture image and exploits the depth-dependent disparity (modeled by $\mathbf{B}_k$) to relate the sub-aperture images for gathering enough information to reconstruct the scene computationally. Refocusing is achieved by varying the depth-dependent disparity model $\mathbf{B}_k$. **b** Seeing through severe occlusions by CLIP as a camera array, with each camera only recording partial nonlocal information of the scene. A obscured object (from the camera with black rays) remains partially visible to some other views (with green rays), whose nonlocal and complementary information enables compressive retrieval of the object. **c** Illustration of instantaneous compressibility of the time-of-flight measurements for a 3D scene in a flash LiDAR setup, where a transient illumination and measurement slice the crowded 3D scene along the depth (time) direction into a sequence of simpler instantaneous 2D images. **d**–**f** CLIP embodiments that directly perform nonlocal image acquisitions with a single-pixel, a linear array, and 2D area detectors, respectively. A single pixel utilizes a defocused spherical lens to integrate a coded image, with $u$ and $v$ behind the lnes being the angular dimension. A cylindrical lens yields along its invariant axis a radon transformation of the en-face image onto a 1D sensor. The complex-valued mask such as a random lens produces a random, wide-field PSF that varies with object depth to allow light field imaging. PSF point spread function, CLIP compact light field photography, LiDAR light detection and ranging, 1D, 2D, 3D one, two, and three-dimensional.

acquired from $l$ views now becomes:

$$\mathbf{f} = \begin{bmatrix} \mathbf{f}_1 \\ \mathbf{f}_2 \\ \vdots \\ \mathbf{f}_l \end{bmatrix} = \begin{bmatrix} \mathbf{A}_1 & \cdots & \mathbf{0} \\ \vdots & \mathbf{A}_2 & \vdots \\ \mathbf{0} & \ddots & \mathbf{0} \\ \mathbf{0} & \cdots & \mathbf{A}_l \end{bmatrix} \begin{bmatrix} \mathbf{B}_1\mathbf{h} \\ \mathbf{B}_2\mathbf{h} \\ \vdots \\ \mathbf{B}_l\mathbf{h} \end{bmatrix} + \sigma = \begin{bmatrix} \mathbf{A}_1\mathbf{B}_1 \\ \mathbf{A}_2\mathbf{B}_2 \\ \vdots \\ \mathbf{A}_l\mathbf{B}_l \end{bmatrix} \mathbf{h} + \sigma = \mathbf{F}(\mathbf{d})\mathbf{h} + \sigma,$$

(3)

where $\mathbf{f}_k$ is a vector that contains $m_k$ measurements at view $k$, and the total number of measurements in $\mathbf{f}$ is $m = \sum_{k=1}^{l} m_k$. The whole system model $\mathbf{F}(d)$ becomes a function of the depth $d$, which is

the key to recover the image $h$ with different focal settings—by applying the shearing matrix $\mathbf{B}_k$ (hence $\mathbf{F}(\mathbf{d})$) to depth $d$ as in light field cameras, the reconstructed image will be correspondingly focused thereon. Supplementary Note 2 details the workflow for image refocusing, novel view synthesis, extending the depth of field, and 3D imaging via depth-from-focus. CLIP thus can attain light field imaging (conventionally of a data size $m \times l$) with a measurement data size of only $m$. It is worth noting that further reduction of the measurement data is possible by multiplexing the measurement from all the views onto a single measurement vector: $\mathbf{f}^c = \mathbf{Tf}$, with $\mathbf{T}$ being the integration operator. Supplementary Note 3 illustrates such a design strategy for a CLIP camera using 2D area detectors.

The nonlocal acquisition strategy is pivotal to encode all scene points of the image into each view's smaller sub-measurement vector $\mathbf{f}_k$ (i.e., $m_k \ll N^2$) for attaining an effective angular resolution of $l$. This is similar to the incoherent multiplexing requirement in compressive sensing, where a rich pool of nonlocal acquisition schemes has been developed for a range of applications during the past decades, benefiting CLIP. Such a nonlocal acquisition also endows CLIP with imaging robustness against defective pixels or scene occlusions. Because the complete scene is encoded in any subset of the measurements, image recovery is not substantially affected by a fraction of defective pixel readings, despite that the conditioning of image reconstruction might deteriorate (Supplementary Note 9). Similarly, an object that is completely blocked in certain views by its surrounding objects, as in Fig. 1b, could be partially visible to the remaining views, which contain incomplete but complementarily global information of the object to enable its retrieval (Supplementary Note 10). Furthermore, CLIP's nonlocal acquisition can take advantage of the compressibility of natural photographs for compressive imaging ($m < N^2$) to minimize data load, particularly when coupled with time-of-flight imaging. As illustrated in Fig. 1c for a point-scanning-based LiDAR imaging of a crowded office scene, the camera captures only a thin slice of the 3D volume at each time bin under an ultrashort illumination. As a result, the crowded 3D scene is decomposed into a sequence of instantaneous 2D images that are far simpler than its continuously-wave-illuminated photograph. Such instantaneous compressibility also holds for NLOS imaging, albeit in a different representation basis[12].

Three exemplar CLIP embodiments utilizing a single pixel element (0D), a linear array (1D), and a 2D area detector are illustrated in Fig. 1d–f, respectively. The single-pixel camera[34,35] (Fig. 1d) sequentially encrypts the scene with different random codes and measures light intensities with a bucket detector. To sample light fields without redundant data, CLIP splits the measurements by scanning the detector during code update along the $u-v$ direction ($u$, $v$, angular axis behind the collection lens) into $l$ positions. With random binary codes, each measurement integrates ~50% of the image pixels, and $m_k \geq 7$ measurements in each view cover every pixel with a high probability ($p = 1 - 0.5^7 > 99\%$). For CLIP with 1D sensors, the x-ray CT imaging model is transformed by using a cylindrical lens to cast along the lens' invariant axis (the axis without optical power) a line-integral of the image onto an individual pixel as in Fig. 1e, allowing a 1D detector array to parallelly records $m_k = N$ measurements to cover all image pixels. Light fields of the scene are then acquired with an array of cylindrical lenses, each being oriented at a distinct angle with respect to the 1D sensor (see the "Methods" section).

For CLIP imaging with 2D detectors (of various sparsity), one can design a complex-valued mask to produce a wide-field, depth-dependent point spread function (PSF) (Fig. 1f) to multiplex sub-aperture measurements (i.e., $\mathbf{f}^c = \mathbf{Tf}$). Moreover, we can unify wavefront coding[36,37], coded-aperture[38] techniques, and diffuser cameras[23,24] into the CLIP framework, where the full recovery of 4D light fields is unnecessary[25]. Detailed formulations and synthetic imaging results for CLIP with 2D detectors, along with additional 1D-sensor-based designs with provably close-to-optimal performances, are provided in Supplementary Notes 3 and 4, respectively. Adaption to camera array systems can be readily accomplished by making each camera (of any dimension) record a few nonlocal coefficients of the scene and sufficiently overlapping individual cameras' fields of view.

We quantified the efficacy of CLIP for light field imaging experimentally with a 0D sensor in Supplementary Note 7, and further evaluated the CLIP reconstruction accuracy synthetically with both 0D and 1D sensors in Supplementary Note 11, which employs CLIP to represent custom-acquired 4D light field data for scenes of different complexities and BRDF characteristics.

**3D imaging through occlusions.** Seeing through occlusions has been previously achieved by dense camera array systems[16], which apply synthetic aperture processing to blur down the occluder while keeping the object of interest coherently enhanced. However, a clear separation of the object and occluder in 3D is difficult due to the defocused background and the limited depth sectioning capacity of camera array systems. We show here background-free 3D vision through severe occlusion with time-of-flight (ToF) CLIP imaging. For the proof-of-concept demonstration, we built a ToF-CLIP system with a streak camera as the 1D ultrafast sensor for snapshot acquisition of large scale 3D time-of-flight data. The streak camera is spatially multiplexed by seven customized plano-convex cylindrical lenslets (diameter of 2 mm and a focal length of 4 mm) along its entrance slit at distinct orientations to mimic a camera array system (see the "Methods" section). The baseline of the camera is 15 mm, and the field of view is 30 mm at a distance of 60 mm. With ~1000 pixels, CLIP implicitly recorded a $125 \times 125 \times 7$ light field dataset and streamed a temporal sequence of 1016 points (i.e., 1000 spatial × 1016 temporal) at a 100 Hz repetition rate for high-speed imaging. A femtosecond laser was modulated by a motorized assembly consisting of a concave lens and a diffuser for programmable illumination between a diverged and a collimated light. The diverged laser shines from an oblique angle to cover both the object and occluder. In practice, an array of synchronized laser diodes is typically employed, with each camera having its own laser source as illustrated in Supplementary Note 10.

Background-free 3D imaging through severe occlusions for three different scenes is shown in Fig. 2a–c. In all cases, the objects that are completely blocked by their preceding items, as rendered in the front view images emulating what a conventional camera would capture, can be well retrieved by the ToF-CLIP camera with correct 3D locations and geometric shapes. For larger objects such as the letter V in Fig. 2b that remain partially visible, its occluded parts are recovered with a weaker intensity. This is because the occluded parts contribute less effective measurement signals for the image reconstruction, equivalent to imaging with a smaller synthetic aperture. Trackings through occlusions is demonstrated in Fig. 2d, where a $2 \times 2$ grid pattern made of white foam was mounted on a translation stage behind a rectangular obscurer and moved back and forth across the camera field of view. Motion of the grid pattern varied the severity of occlusion smoothly from none to a complete obscurance as shown in the representative frames. Except for a weaker intensity caused by occlusion, the grid pattern is adequately recovered at all the time instances. The complete video of tracking through occlusion is provided in Supplementary Movie 1 along with reference photographs.

It is noteworthy that a clear separation between the objects and occluder is consistently achieved in all the scenes. No defocused background signals from the occluder are discernible on the blocked object, highlighting the benefits of merging dense multi-view measurement with ToF by CLIP. Because an occluder reduces the number of measurements for the blocked object, and the current CLIP camera has a compression factor of ~20 (with respect to a single sub-aperture image), the occluded objects that can be well recovered are restricted to be relatively simple in geometry. Nonetheless, the imaging outcomes are still remarkable, considering the reduction of measurements against camera

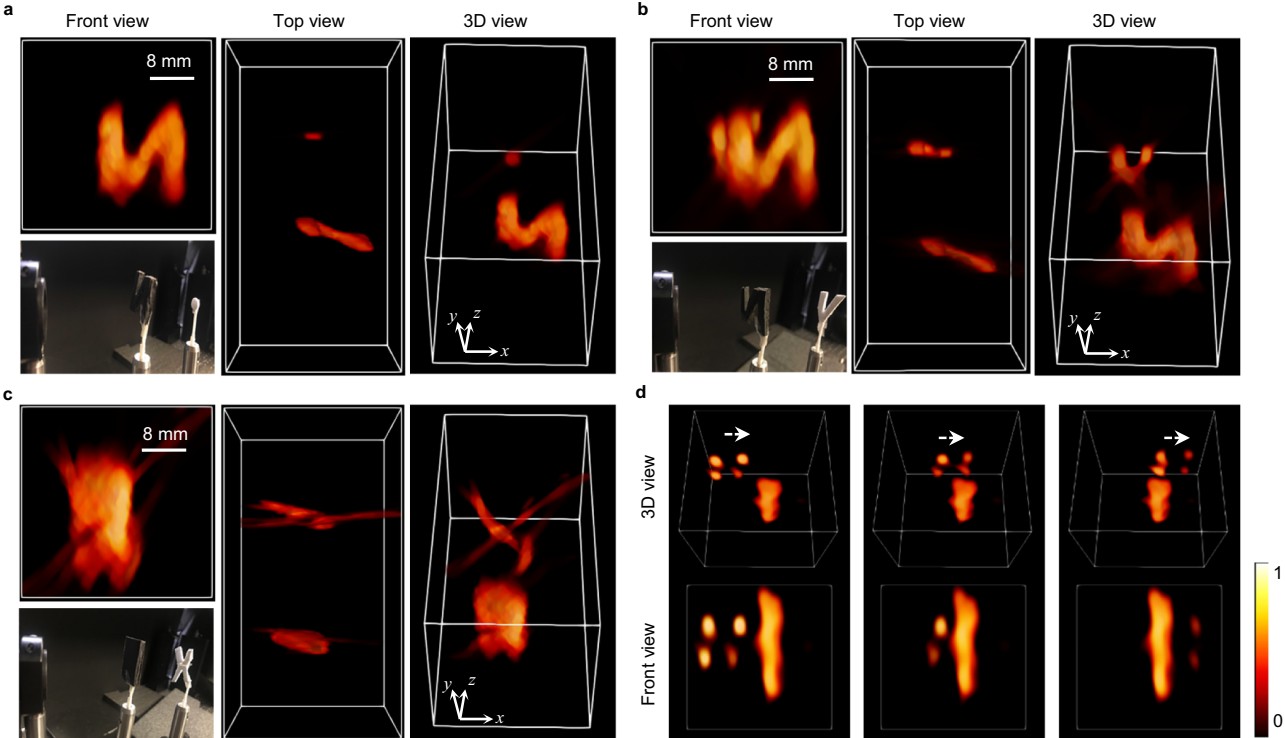

**Fig. 2 Three-dimensional imaging (3D) through occlusions. a–c** Reconstructed 3D images rendered in different perspective for three scenes: circular plate (**a**) and letter V (**b**) behind the letter N, and letter X (**c**) blocked by a rectangular plate. The severe occlusions are evident from the front view images, with the larger objects in the front completely blocked the object right behind them. In contrast, CLIP is able to unambiguously reconstruct the obstructed objects in 3D without any defocusing signals from the preceding occluder. **d** Three representative frames of imaging a 2 × 2 grid pattern moving across the CLIP camera FOV behind a rectangular occluder. Note that signals from the black occluders are enhanced relative to the objects for better visualization.

array systems of the same resolution is more than 100 folds, and conventional compressive imaging using the same amount of data shows similar imaging characteristics but lacks light field capability to see through occlusions. We further quantified the accuracy of CLIP imaging through occlusions via synthetic studies in Supplementary Note 10, which shows a small imaging error (<10%) can be obtained by CLIP despite of a large reduction (>100 times) in light field measurement data.

**Flash LiDAR within an extended depth range**. For high-quality 3D imaging of indoor scenes, multi-view methods require an unwieldy system baseline apart from their dependence on object texture. By contrast, flash LiDAR imaging can maintain a high precision at longer distances in a compact form but suffers from a stringent tradeoff between the sensing range and light throughput[15,39]. To demonstrate that CLIP is well-posed to lift such a tradeoff, we tuned the camera's field of view to 1.5 m × 1.5 m at a nominal distance of 3.0 m by moving the lenslet array closer to the slit and aligned the laser to be approximately confocal with the camera while providing a diverged illumination after passing through the motorized assembly.

An example of single-shot flash LiDAR imaging with an extended depth range is shown in Fig. 3a, where several texture-less letters were placed at different depths spanning a range of ~2 m. The extended depth of field is highlighted in Fig. 3b by computationally refocusing the camera from far to near, as indicated in the top view image. The resultant LiDAR projection photograph clearly renders the defocusing blur for objects that deviate from their actual focal settings, whereas an all-in-focus image generated by CLIP in Fig. 3c allows a sharper portrait of the entire 3D scene. The flash LiDAR imaging resolution was estimated to be about 30 mm laterally and ~10 mm axially (depth

direction). While this example features a relatively simple scene to facilitate the comparison between the reference photograph and LiDAR images, additional results for handling more complex scenes are presented in Supplementary Note 14.

We demonstrated ToF-CLIP in dynamic imaging of a 3D scene by mounting a letter V on a rotation stage and manually rotating it at an irregular speed against a simple and cluttered background, respectively. The resultant motions were filmed by the CLIP-ToF camera at a 100 Hz frame rate and a reference video camera at 60 Hz. The two videos were then numerically synchronized after temporally downsampling the LiDAR results to 60 Hz for comparison. Representative frames of the dynamic results are shown in Fig. 3d and e for the simple and cluttered background, respectively, and the full comparison videos are provided in Supplementary Movies 2 and 3. To better visualize the rotating dynamics of the letter V in the cluttered scene, we isolated it in a magnified view in the projection image in Fig. 3e. CLIP captured the motion of the letter faithfully in both LiDAR videos and reproduced the letter's small vibrations during rotation. We also recovered the unobstructed hand in the simple scene and the dynamic shadows cast by the letter V on the background walls in the cluttered scene, despite of compressive data acquisition in the current ToF-CLIP camera.

**NLOS imaging with curved and disconnected surfaces**. Unlike LiDAR that detects the directly scattered photons, NLOS imaging analyzes multiply scattered light from a diffusive surface to reveal objects hidden from direct line-of-sight. A key ingredient for such an analysis is the precise knowledge of the relaying surface's 3D geometry, which was previously obtained via nontrivial calibrations by a stereo[26] camera or scanning-based LiDAR[27,40] ranger, hampering applications in the field where the relay surface

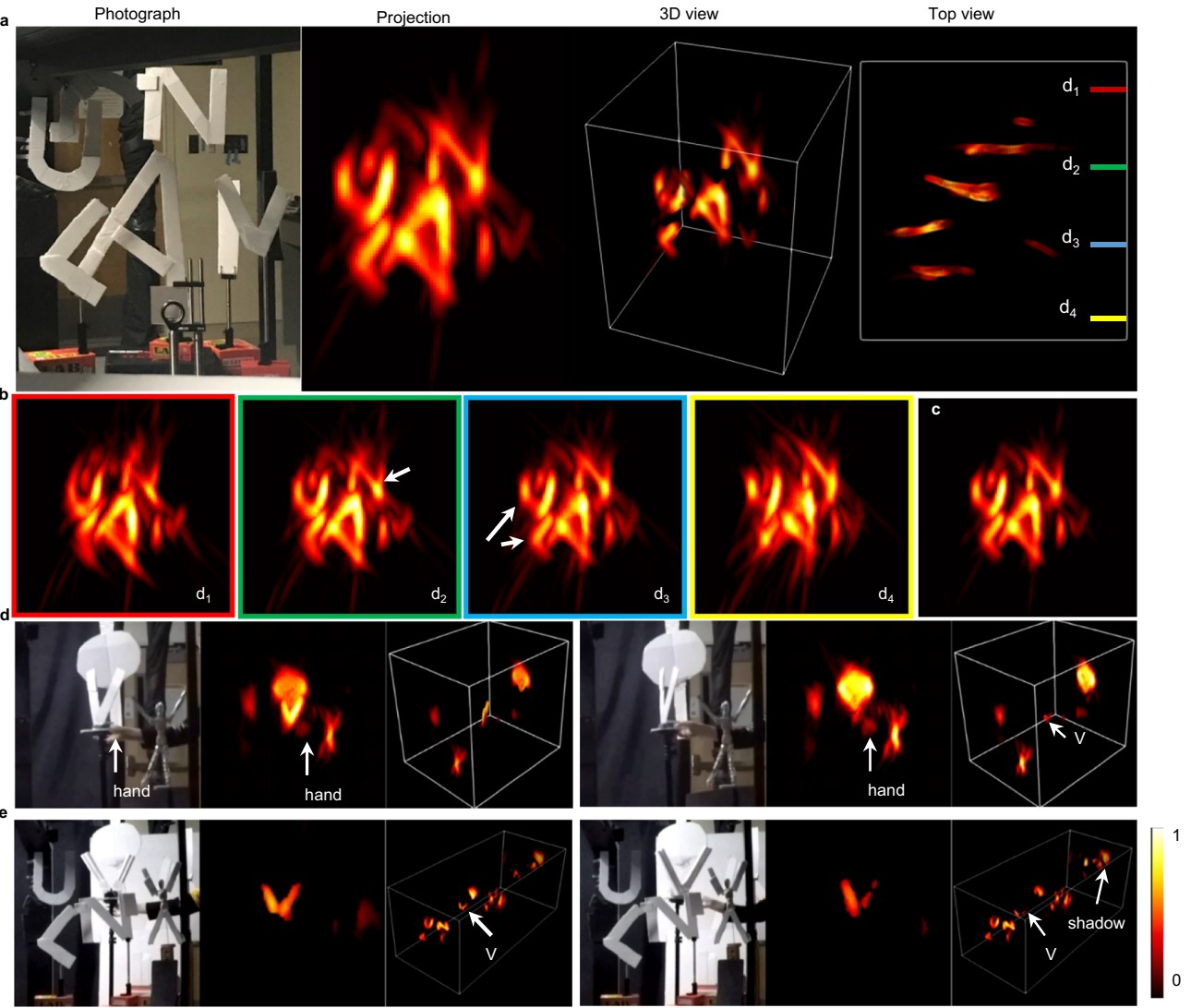

**Fig. 3 Snapshot flash LiDAR imaging over an extended depth range. a** Flash LiDAR imaging of a letter scene. From left to right are the reference photographs, a projected two-dimensional LiDAR images along the depth direction, and the 3D (three-dimensional) point-cloud representation of the scene. **b** flash LiDAR of the same 3D scene without extending the imaging depth of field, obtained by refocusing the camera onto a single focal plane. Note the defocus blur in the near and far objects. **c** Computational all-in-focus image. **d** and **e** Two representative frames for the dynamic imaging of a manually rotated letter V in a simple and cluttered scene, respectively.

evolves with the camera's viewpoint and 3D surroundings. The ToF-CLIP camera addresses this critical need for real-time mapping of the relay surface via built-in flash LiDAR imaging. More importantly, it can accommodate a non-planar surface geometry for NLOS imaging using array detectors with its light field capability. Paired with a proposed hybrid time-frequency domain reconstruction algorithm, which can handle general surfaces with a computational complexity of $o(N^4)$ (see the "Methods" section), ToF-CLIP can attain real-time NLOS imaging with arbitrary curved surfaces. While NLOS imaging with a dynamic and curved surface has been demonstrated by Manna et al. [40], its reception point was fixed at a stationary point rather than being on the dynamic surface, making it inapplicable for real-time imaging with array detectors. Similarly, the preprocessing step [41] proposed by Lindell et al. that adapts the $f{-}k$ migration reconstruction algorithm to deal with slightly curved surfaces in confocal NLOS imaging has a computational complexity of $o(N^5 \log N)$, which is higher than the time-domain phasor field method and thus inefficient for real-time reconstruction.

To demonstrate our approach, we directed the CLIP-ToF camera towards a scattering wall with a fixed focus. The field of view is tuned to be ~0.5 m × 0.5 m at a standoff distance of ~1 m. The geometry of the wall was mapped by the flash LiDAR, and NLOS signal reception was delayed accordingly to avoid the strong reflections from the collimated laser spot on the wall. The hidden scene was then reconstructed in real-time by the hybrid time–frequency domain algorithm.

We demonstrated NLOS imaging with planar, disconnected, and curved walls in Fig. 4a–c. For all the relay walls, the hidden scenes were placed over 1 m away from the laser spot on the wall and then imaged with a single laser shot at an average laser power of 700 mW. The 3D flash LiDAR measurement of the walls are shown in the first column, and the NLOS imaging results for two example objects in each category were rendered in a 2D front view (from the wall's perspective) and a 3D point cloud format in the following columns. Both the 3D positions and morphological structures of the hidden objects were decently recovered for NLOS imaging with all the relay walls. The importance of an extended depth-of-field to cope with disconnected and curved

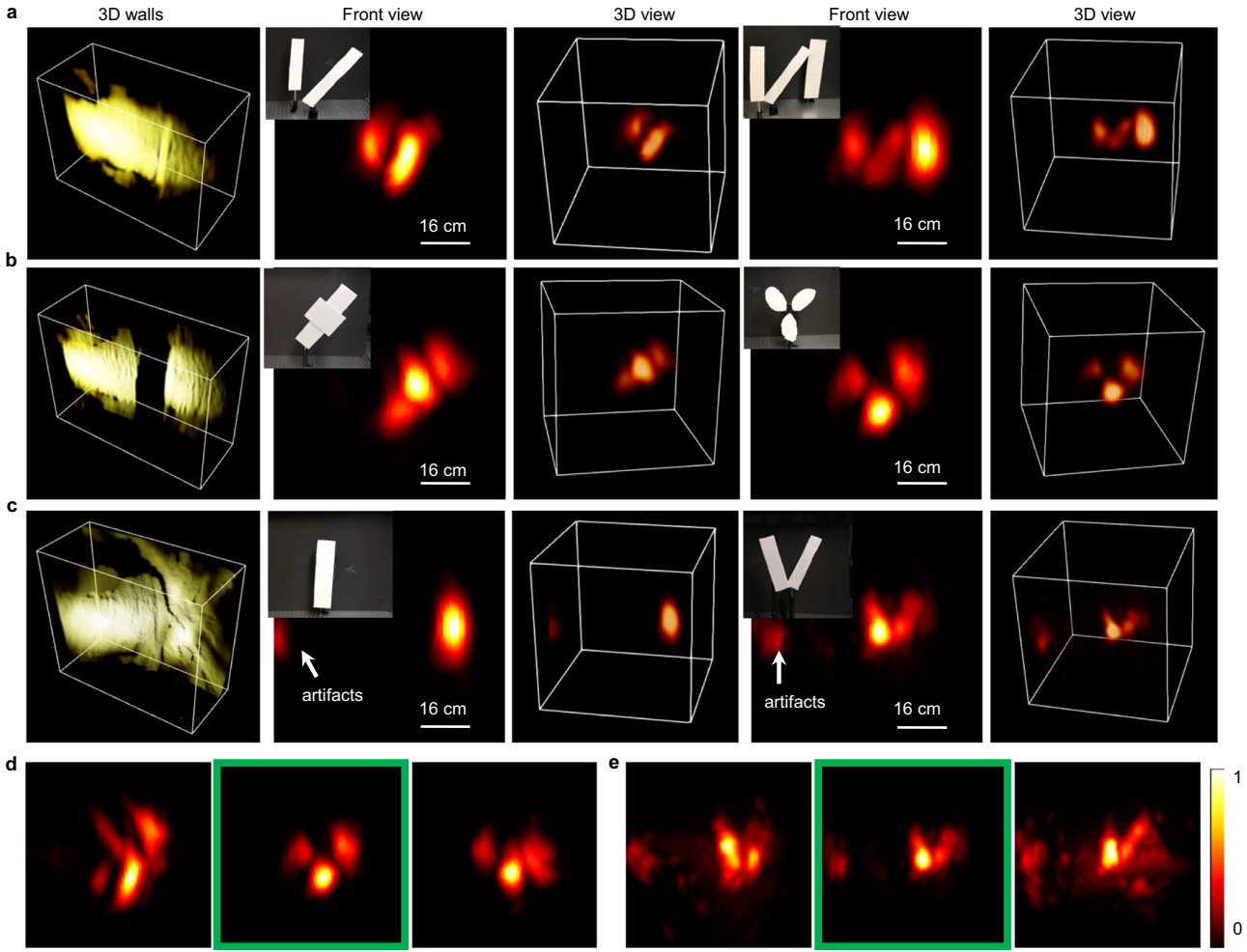

**Fig. 4 NLOS imaging by CLIP-ToF. a–c** Imaging with planar, disconnected, and curved surfaces, respectively. From left to right are the flash LiDAR imaging of the relay surfaces, and two example hidden objects rendered as a projection image in the front view, and a 3D (three-dimensional) point cloud. Ground truth photographs of the object are shown in the inset of the front view image. **d, e** Reconstructed NLOS images for the disconnected and curved surfaces, respectively, with defocus errors on the relay wall, and those recovered with extended depth of field (highlighted by the green box). The quality of reconstruction degrades when the camera's extended depth of field is disabled.

surfaces is illustrated in Fig. 4d, e, respectively, where the camera's extended depth of field is disabled by computationally refocusing the camera onto different planes (from rear to the front) before reconstructing the hidden scenes. Due to defocus effects that blur the spatiotemporal data on the walls, the reconstruction quality degrades noticeably compared with the images reconstructed with an extended depth of field (highlighted by the green boxes). It is worth noting that NLOS imaging with the curved surface suffered from secondary laser inter-reflections (i.e., laser reflections between the surface parts before incident onto the hidden objects) during the experiments, which caused the imaging artifacts in Fig. 4c, despite that the phasor field method is robust against multiple inter-reflections. This is primarily because the secondary laser reflection is much stronger than the inter-reflections of the weaker NLOS photons. Still, CLIP's capability to handle disconnected and curved surfaces is an important step to achieve point-and-shoot NLOS imaging in the field.

## Discussion

The angular and temporal axes are two basic elements in the plenoptic function $P$(spatial: $x$, $y$, angular: $u$, $v$, wavelength: $\lambda$, time: $t$), which completely characterizes the fundamental properties of light. With decades' investigations and developments, current 3D imagers have reached the practical sensing limit about what is offered by measuring along the angular or temporal axis. A more complete acquisition of the plenoptic function (or the light fields), such as both angular and temporal dimension, towards versatile and more capable 3D vision remains a largely untapped area. This is mainly attributed to the lack of efficient schemes for sampling the resultant high-dimensional data with a 2D image sensor. However, for most applications, recording the entire high-dimensional light field is not the ultimate goal but an intermediate step to gain versatile image processing abilities, such as digital refocusing, extending the depth of field, or depth extraction. At the same time, expanding to the high-dimensional space promotes data sparsity[12], making full-rate acquisition unnecessary and inefficient. Previous endeavors to sample light fields in an implicit and more efficient manner include the coded aperture[42] and wavefront-coding[36,37] techniques. Unfortunately, their acquisition schemes for 3D imaging[37] or depth of field extension[36] require a densely sampled 2D image (Supplementary Note 3) that precludes their applications in ultrafast, infrared[34,35], or Terahertz[43] imaging applications, where the detector resolution is severely limited. The CLIP framework encompasses and goes far beyond these methods to utilize sensors of arbitrary formats for efficient light field imaging (compressive

or not), with a flexible nonlocal sampling strategy that promotes imaging robustness and better exploitation of the sparsity characteristic of high-dimensional data. An important application of canonical light field camera that remains to be explored by CLIP is to measure, and consequently correct for, optical aberrations as it can recover the full 4D light fields (see Supplementary Note 5). Still, even without recovering the 4D light field, CLIP can readily correct for the Petzval field curvature owing to its refocusing capability, which can facilitate the coupling of planar sensors with monocentric systems[44,45] for wide-field or panoramic imaging.

CLIP's recording of a 4D light field is essentially an efficient dimensionality reduction in the optical domain, allowing high-dimensional information to be acquired with sensors of lower dimensionality such as the ubiquitous 1D or 0D (single pixel) detectors, which are still the dominant sensor format for imaging at the ultrafast time scale or the infrared, terahertz spectral regime. This feature facilitates CLIP to be deployed as a universal platform for snapshot multidimensional imaging[46] that samples the plenoptic function in a massively parallel manner. For instance, by extending the 1D ultrafast sensor in CLIP to 2D area detectors such as a megapixel SPAD, the extra spatial dimension could be readily used to measure the objects' spectra. Quantitative polarization information can also be extracted by overlaying the pixels with a layer of polarizers, similar to utilizing a color filter array for color photography. The polarization cues have been routinely used for depth sensing and refinement[47,48], which can potentially further bolster CLIP's 3D vision performance.

## Methods

**Experimental setup**. A high dynamic range streak camera (C13410-01A, Hamamatsu Photonics) is used as the 1D array of time-of-flight sensors, which parallelly record the temporal signal within a single snapshot and a temporal resolution around 60 ps (for an observation window of 10 ns). For CLIP implementation with 1D sensor, seven customized plano-convex cylindrical lenslets (diameter of 2 mm and a focal length of 4 mm) are secured on a 3D printed holder and mounted on 3-axis translation stage to align them with the streak camera's entrance slit, which is effectively the virtual sensor plane. The cylindrical lenslets are rotated to different angles that are approximately uniform in the range of [−45°, 45°] to obtain incoherent measurements among different views. For optimal imaging through occlusions, the cylindrical lenslet angles are further randomly distributed along the lenslet array direction, as the effective measurement entries for the occluded objects are reduced to a subset of the measurement entry in the imaging model. Such random distribution maximizes statistically the incoherence among any subset of the measurements to ensure consistent image recovery performance. The imaging field of view is adjusted by tuning the translation stage that changes the distance between the lenslets and streak camera's slit. For transient illumintion, a femtosecond laser (808 nm, 7 mJ per pulse, ~100 fs pulse width, Astrella-F-1K, Coherent Inc.) is synchronized with the streak camera and trigged at a repetition rate of 100 Hz and an average power of 700 mW. An assembly comprised with a concave lens (LD1464-B, Thorlabs) and a diffuser (EDC-20-14175-A, RPC Photonics) is switched by a step motor to diverge the laser beam for flooded illumination in flash LiDAR imaging. For NLOS imaging, the assembly is displaced for a collimated radiation.

**Image reconstruction**. Recovering a 4D light field or a refocused image from Eqs. (2) and (3) can be inverted by solving a corresponding optimization problem:

$$\mathrm{argmin}_{\hat{h}}||\mathbf{f} - \mathbf{F}(\mathbf{d})\mathbf{h}||_2^2 + \mu||\varphi(\mathbf{h})||_1, \quad (4)$$

$$\mathrm{argmin}_{\hat{\mathbf{P}}}||\mathbf{f} - \mathbf{A}'\mathbf{P}||_2^2 + \mu||\varphi(\mathbf{P})||_1, \quad (5)$$

where $|| \cdot ||_1$ is the $l_1$ norm and $\varphi(\cdot)$ is a domain in which the 4D light field $\mathbf{P}$ (or image) is sparse. $\mu$ is a hyperparameter that balances the data fidelity and regularization term. In the framework of regularization by denoising algorithm[49], the representation domain $\varphi(\cdot)$ is not explicitly specified and the regularization step is implemented by a state-of-the-art image denoising algorithm such as BM3D or even a neural network. We adopted the BM3D and total variation (TV) denoisers for the regularization, owing to the existence of efficient algorithms[50]. Also, to minimize light field processing time that involves numerous refocusing steps (depth retrieval, extending depth of field), the reconstruction process at each step is initialized with the previous solution to exploit the proximity of the solutions.

It is worth noting that while recovering the 4D light field is always compressive in CLIP, directly retrieving a refocused image is not necessarily the same. Still, a major appeal of CLIP is to use a small number of sensors for recording a large-scale

light field, which typically falls into the compressive sampling regime. In this case, we show in Supplementary Note 6 that while the imaging model designed in or transformed by CLIP may not satisfy the restricted isometry property (RIP) to guarantee uniform recovery of arbitrary images in the classic sparse signal model, CLIP has the generality in the structured-sparse signal model and hence remains applicable for practical imaging applications.

**Camera calibration**. To obtain quantitative and absolute 3D positions for flash LiDAR imaging, the CLIP camera is metric-calibrated to extract its intrinsic matrix. A calibration pattern is made of a planar blackboard with a grid of $3 \times 4$ diffusing plates (circular white foam with a 10 mm diameter). With a femtosecond laser illumination, the CLIP camera captures the calibration pattern at ~10 different orientations, and the grid positions are automatically extracted in each image. The intrinsic matrix of the camera is then obtained by Zhang's calibration method[51].

**Flash LiDAR and NLOS experiments**. Room light was turned on during all experiments and the gating functionality of streak camera was enabled to makes it robust against ambient illuminations for time-of-flight imaging.

*Flash LiDAR coordinate transformation*. With the camera being the origin (assuming roughly confocal illumination and detection), the flash LiDAR produces distance measurements in a polar coordinate, not the direct $z$ components. The radial distance along different imaging pixels needs to be transformed into a rectilinear coordinate for correct 3D modeling. Given the camera's intrinsic matrix $K$ and the homogeneous pixel coordinate $[u, v, 1]$, each pixel's projection angle with respect to the camera's optical axis can be derived as

$$\begin{bmatrix} \tan\theta_x \\ \tan\theta_y \\ 1 \end{bmatrix} = \begin{bmatrix} x/z \\ y/z \\ 1 \end{bmatrix} = K^{-1}\begin{bmatrix} u \\ v \\ 1 \end{bmatrix}, \quad (6)$$

where the absolute position of a point object is denoted as $[x, y, z]$ in the rectilinear coordinate system. The LiDAR measurement distance $r$ is related to the rectilinear coordinate as

$$r = \sqrt{x^2 + y^2 + z^2} = z\sqrt{(x/z)^2 + (y/z)^2 + 1}. \quad (7)$$

Combing Eqs. (6) and (7), the absolute 3D position in the rectilinear coordinate is obtained as

$$\begin{bmatrix} x \\ y \\ z \end{bmatrix} = z\begin{bmatrix} x/z \\ y/z \\ 1 \end{bmatrix} = \frac{r}{\sqrt{(x/z)^2 + (y/z)^2 + 1}}K^{-1}\begin{bmatrix} u \\ v \\ 1 \end{bmatrix}. \quad (8)$$

*Time-gain compensation of flash LiDAR signal*. Unlike conventional scanning-based LiDAR that employs collimated laser illumination, the diverging light in flash LiDAR has a $r^2$ intensity decay, which makes distant objects to receive (and reflect) much fewer photons. We employed a time-gain compensation $(t-d)^2$ to partially equalize the signal intensity for objects positioned at a distance larger than $d$ but shorter than $d_2$, after which the signals are too noisy for non-reflective objects. A customized compensation curve could also be used for the best visualization of the 3D scene, a common practice in medical ultrasound imaging.

*NLOS geometric calibration*. The absolute 3D coordinates of the relay surface in NLOS imaging are measured via flash LiDAR in real-time. To extract the laser spot position on the wall and to cope with different walls in field applications that will modify the laser position thereon, the fixed propagation line of the laser is characterized in the absolute 3D space. The laser illumination spot on the surface is then dynamically calculated by intersecting the propagation line with the relay wall's 3D point cloud. To parameterize the propagation line, two thin microscopic slides are placed along the collimated beam path, with the tiny Fresnel reflection from the slides encoding two points that the laser passed through. The absolute positions of the two points are then measured using the LiDAR technique.

*Hybrid frequency-time domain NLOS reconstruction*. Real-time NLOS imaging with arbitrarily-curved surfaces lacks an efficient solver. Recent developments of fast reconstruction algorithms[27,41,52] significantly reduced the computational cost to make real-time imaging feasible, but are exclusively limited to the paradigm of approximately planar surfaces. The pre-processing method[41] presented by Lindell et. al. can transform the measurement data from a curved surface into a format suitable for fast frequency-domain solvers, but has a computational complexity of $o(N^5 \log N)$, which is slightly more expensive than the universal filtered back-projection algorithm and thus less appealing. Time-domain methods are more flexible in accommodating arbitrary wall geometries but more time-consuming. The hybrid frequency–time domain reconstruction method proposed here first converts the spatiotemporal measurement on a curved surface $y_r(\mathbf{r_p}, t)$ onto a virtual plane via wave propagation in time domain and then reconstruct the hidden scenes with existing efficient frequency-domain phasor field method[52]. Under the phasor-field framework, the spatiotemporal waveform on the virtual plane can be

**Table 1 Comparison of NLOS imaging computation and memory complexity.**

| Algorithms | Computational complexity | Memory complexity | Execution time | | |
|---|---|---|---|---|---|
| | | | CLIP reconstruction (s) | NLOS reconstruction (s) | Total reconstruction (s) |
| $f$-$k$ migration | $o(N^5 \log N) + o(N^3 \log N)$ | $\sim o(50N^3)$ | 0 | 15.1+ 0.65 (CPU) | 15.7 (CPU) |
| Time-domain phasor field | $o(N^5)$ | $o(N^3)$ | 0 | 0.40 (GPU) | 0.40 (GPU) |
| Hybrid time–frequency method | $o(N^4)$ | $o(N^3)$ | 2.0 or 0.01 (adjoint method) | 0.03 (GPU) | 2.03 or 0.04 (GPU) |

calculated in time-domain as

$$f\left(r_v, t\right) = \int_{-w}^{w} y_r(\boldsymbol{r_p}, t) * p(t-\tau) \mathrm{d}\boldsymbol{r_p}, \qquad (9)$$

where $\tau = \frac{r_v - r_p}{c}$ is the travel time from the point $\boldsymbol{r_p}$ on the curved surface to a point $\boldsymbol{r_v}$ on the virtual planar surface, and $p(t)$ is the convolutional kernel in the phase-field method. The detailed geometry is included in Supplementary Note 12. Wave migration in the time domain has two major advantages. First, it is more efficient than frequency domain migration: with both the curved and virtual plane being sampled with a spatial resolution of $N^2$, time-domain migration has a computational complexity of $o(N^4)$ instead of $o(N^5 \log N)$. Second, it does not restrict the sampling pattern on either the curved surface or the virtual plane. While the frequency-domain method needs a spatial interpolation operation to deal with nonuniform sampling on a planar surface due to a 2D camera's projective distortion, time-domain migration can readily achieve a regular sampling on the virtual plane for the subsequent frequency-domain reconstruction[52]. Combined with the complexity of $o(N^3 \log N)$ for the frequency-domain phasor field reconstruction, the total complexity for the hybrid time-frequency domain reconstruction is $o(N^4)$, still orders of magnitude faster than time domain methods. The memory complexity for the frequency-domain phasor field and relevant $f$-$k$ migration reconstruction have been analyzed in the literature to be $o(N^3)$ and $\sim o(50N^3)$[52], respectively. The time-domain migration in Eq. (9) has a memory complexity of $o(N^3)$ in order to store the propagated signals $f\left(r_v, t\right)$ on a virtual plane, leading to a total complexity of $o(N^3)$ for the hybrid time–frequency domain reconstruction method.

We further accelerate the reconstruction on a GPU (Nvidia RTX3080Ti) using CUDA. For a $128 \times 128 \times 128$ imaging volume with a spatiotemporal data cube of $125 \times 125 \times 1016$, the NLOS reconstruction time is $\sim 0.03$ s, which can reach a 30 Hz video rate. The actual bottleneck lies in the iterative CLIP reconstruction of the spatiotemporal data cube on the wall, which takes about 2.0 s. However, we show in Supplementary Note 13 that a fast CLIP solution via the adjoint operator can reduce the reconstruction time to 0.01 seconds for NLOS imaging at the expense of noise robustness. Table 1 summarizes the computation and memory complexity of the hybrid time–frequency domain reconstruction method against the time-domain phasor field method and $f$-$k$ migration for NLOS imaging with curved surfaces. It is noteworthy that the complexity of $f$-$k$ migration includes the necessary preprocessing step for coping with curved surfaces, and its execution time is obtained by CPU processing with a downsampled spatiotemporal data cube of $(32 \times 32 \times 512)$ instead of $(125 \times 125 \times 1016)$. Conforming with the complexity analysis, the preprocessing step is more time consuming than the actual reconstruction in $f$–$k$ migration.

## Data availability
The data of this study is available at https://github.com/fengxiaohua102/CLIP.

## Code availability
The code of this study is available at https://github.com/fengxiaohua102/CLIP.

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

## Acknowledgements

This work is supported by the National Institute of Health (NIH/NIGMS) under the grant number of R35GM128761(L.G.).

## Author contributions

X.F., Y.M. and L.G. conceived the study. X.F. designed the imaging method, wrote the reconstruction code, and analyzed the data. Y.M. built the imaging systems, performed the experiments, and analyzed the data. All authors contributed to the manuscript preparation. X.F. and L.G. supervised the project.

## Competing interests

The authors declare no competing interests.
