## [Peer Review File · Nature Communications]

Compact light field photography towards versatile three-dimensional visionEditorial Note: Parts of this Peer Review File have been redacted as indicated to remove third-party material where no permission to publish could be obtained.

REVIEWER COMMENTS

Reviewer #1 (Remarks to the Author):

This paper describes a method for reconstructing scene images from a light field captured by a camera array that relies on sparse sampling of the camera images. This adjustment results in faster and more robust reconstruction method. The authors demonstrate their method using several experimental implementation and application examples. They also introduce a new method for NLOS image reconstruction from non-planar relay surfaces. The paper does not seem to mention that this problem has been addressed in other works, even though the papers are cited.

Overall the paper is well written and introduces an interesting method. However there are several issues that need to be clarified before it can be evaluated.

1. Some of the intro sounds a little like a sales pitch. For example lines 70 to 74. Maybe try to use some more objective language here.
2. It would be good to highlight the difference between the proposed reconstruction model and prior compressive light field approaches. Both propose a linear optimization method, and include a perspective transform (i.e. shearing) to do the reconstruction so the methods seem very similar. Is the main difference between the papers the proposed capture hardware?
3. The paper uses algorithm speed as a major factor to motivate the work, but doesn't do a lot to quantify or verify that statement. The authors mention runtimes of methods in some places, but often the runtimes include a bulk time for different methods like line of sight and NLOS reconstruction. There is also no comparison of computational or memory complexity. I think the paper needs to include some meaning full comparison to alternative methods and a discussion of the expected improvements in performance over prior methods. It also needs to provide complete and structured information about the actual execution speeds and put those in some meaningful context.
4. Similarly, the paper talks about the robustness of the work to missing or erroneous pixels, but does not actually do anything to test that.
5. I'm a little confused about the actual setup used in the different demonstrations. The imaging setup with the streak camera and lenslet array should result in a light field array with a diameter similar to the slit of the streak camera. So a centimeter or two. An array of that size should not be big enough to image around the occlusions they create. The authors explain the geometry that determines the permissible size of the occluder in the supplement. I think some added clarification and maybe a sketch of the setup is needed here.
6. NLOS imaging using non-planar and changing relay walls has been demonstrated by La Manna et. al. (43). The method presented here is probably faster, but a little more discussion of the methods is probably necessary. The paper about the FK migration algorithm that the authors use also describes reconstruction from non-planar surfaces.
7. The computationally most challenging step of FK migration is the necessary interpolation in the Fourier domain. That is what drives complexity and memory use and tends to create artifacts. The published code the authors use addresses this problem by oversampling the reconstruction in the Fourier domain resulting in very high memory use. In that light, the statement, that FK does not require interpolation is misleading. The authors refer to interpolation in the time domain. FK instead needs interpolation in the Fourier domain.

Reviewer #2 (Remarks to the Author):

The manuscript reports on a method for lightfield photography in which, in essence, instead of acquiring L different images from different view points, only one pixel or one line of each image is acquired but each one from a different perspective. These are then combined together through an minimisation approach that relies on a "shear" operator that models how the various parts of the scene are captured at varying view points and then registered in a single final image.

The idea is clever and seems to deliver very promising results. The authors show many different possible implementations of the technique, ranging from 2D imaging, flash lidar to non line of sight imaging.

I am not personally convinced that the NLOS imaging results are that significant compared to the state of the art. However, the other results do look convincing, including the measurements in the presence of occluders. The video material provided is also very convincing.

The work is very carefully prepared with sufficient details to reproduce the results. My only comment is that the supplementary information is actually very much integral to the main work as many or most of the actual results are presented there. This is probably a choice based on the fact that the authors present so many different cases that it is hard to show all results in the main text. But this is just a stylistic choice and does not impact the importance of the work itself.

I therefore suggest acceptance of this work for publication without any need for revision.

Reviewer #3 (Remarks to the Author):

In this manuscript, the authors report their development of an imaging method which they call "compact light field photography (CLIP)". They claim that CLIP enables three-dimensional imaging with fewer detectors compared to conventional light field photography methods. They demonstrated volumetric imaging by combining CLIP with other imaging techniques, such as time-of-flight, LiDAR, and non-line-of-sight 3D imaging. The main argument of this work is that the data size can be reduced compared to conventional light field photography, which is advantageous for large-scale, high-dimensional photography. However, I do not think the quality of the manuscript meets the publication criteria of Nature Communications in terms of novelty, quality of presentation, and impact of the results. Detailed comments are listed below:

1. I doubt the effectiveness of their approach toward the realization of dynamic 3D imaging. The authors perform data compression optically by using specially arranged sensors (a single pixel, a linear array, or a sparse 2D area detector). Whether this compression works for retrieval of 3D images depends on the scene (as long as the restricted isometric property of the measurement matrix is not evaluated). The authors' approach seems to inherently lack generality.
2. Also, optical compression accompanies the loss of data due to difficult-to-control factors. For example, in the authors' setup in Fig. S5d, the angles and positions of cylindrical lenses, the widths of slits, and the distance between the lenses and the sensor critically change the intensity profile on the sensor. In addition, aberration due to the imperfection of the lenses induces loss of information. As long as one can acquire the entire image, it should be taken. If compression is needed, we can do it by post processing using an FPGA in a high-throughput, lossless, and reproducible manner.
3. The definition of CLIP is unclear. The authors' statement "To address these challenges, we present compact light field photography (CLIP) to sample dense light fields with a drastically improved efficiency and flexibility. By employing nonlocal image acquisitions and distributing a complete acquisition process into different views, CLIP enables light field imaging with a measurement dataset smaller than a single sub-aperture image and remains natively applicable to camera array systems"

sounds no more than compressed light field photography, which has been thoroughly investigated.

4. Even with the authors' explanation, "Unlike previous compressive light field cameras^{23–25} that decode a densely sampled 2D image into a full 4D light field, CLIP features the unique capability of utilizing a small number of sensors arranged in arbitrary formats—a single pixel, a linear array, or a sparse 2D area detector—for light field imaging", the difference is unclear because one can easily reduce the effective number of datapoints on a CCD /CMOS camera by pixel binning.

5. The performance of their method is not evaluated well. In compressed sensing, evaluation of data fidelity is essential. Without comparison of the reconstructed images with the ground truth measured by conventional methods (with a lower acquisition rate), it is impossible to judge if the method is good or not.

6. The authors' main claim "enable snapshot 3D imaging with an extended depth range and through severe scene occlusions" in the abstract is suspicious. In the supplementary movies, the shape of the objects significantly changes when they are occluded. Again, quantitative evaluation image reconstruction is needed for supporting their claim.

Response Letter

The authors thank the reviewers for their insightful comments, which have greatly improved the quality of our manuscript.

For a brief summary, the major revisions on the manuscript are as follows:

1. We compare with previous compressive light field imaging methods with great details to show that CLIP is a unique framework to design and transform any imaging methods that employs nonlocal data acquisition into an efficient light field imaging method. CLIP is further shown, via experimental and synthetic studies, to be capable of recovering the 4D light field or directly retrieving a refocused image, and we prove that the later approach has the advantage of accommodating complex scenes better.
2. A comparison on the computation and memory complexity was made between CLIP and existing compressive light field imaging methods. The proposed hybrid time-frequency domain NLOS reconstruction algorithm was also benchmarked against alternative methods for imaging with curved surfaces, including the f-k migration and time-domain phasor field method.
3. The robustness of CLIP against defective sensor readings (missing and dead pixels) are evaluated via both experimental and synthetic studies.
4. We add new experiments to quantify the imaging accuracy of CLIP with 0D sensors, and add extensive synthetic simulations to evaluate the imaging accuracy of CLIP with both 0D and 1D sensors under different sampling regimes (compressive or not).
5. We add explanations and new numerical experiments (the generalized flip test) to prove the generality of CLIP for practical imaging applications, despite that the restricted isometry property (RIP) cannot be directly evaluated (which is a NP hard problem).
6. We improved the new imaging experiments for seeing through occlusions, illustrated the geometry, and further evaluated the accuracy of CLIP for this particular applications via extensive synthetic studies.

In the following, we provide point-by-point responses. The changes in the manuscript are highlighted in red.

Reviewer 1

This paper describes a method for reconstructing scene images from a light field captured by a camera array that relies on sparse sampling of the camera images. This adjustment results in faster and more robust reconstruction method. The authors demonstrate their method using several experimental

implementation and application examples. They also introduce a new method for NLOS image reconstruction from non-planar relay surfaces. The paper does not seem to mention that this problem has been addressed in other works, even though the papers are cited.

Overall the paper is well written and introduces an interesting method. However, there are several issues that need to be clarified before it can be evaluated.

Response:

We thank the reviewer for the constructive comments on this work. The previous NLOS image reconstruction for non-planar surfaces is compared with the proposed method in the revised manuscript. The main limitation of that work is its higher computation and memory complexity, please see Response to Comment 3 and 6 for details on this particular point. We addressed the comments in detail below.

1. Some of the intro sounds a little like a sales pitch. For example lines 70 to 74. Maybe try to use some more objective language here.

Response:

We revised the lines 70-74 to describe the method in a technical and objective flavor, which is appended below.

“ ... To address these challenges, we present compact light field photography (CLIP) to sample dense light fields^{1,2} with a drastically improved efficiency and flexibility. Unlike previous compressive light field cameras³⁻⁵ that require densely sampled 2D images for recovering a 4D light field, CLIP is a systematic framework to design and transform any imaging model that employs nonlocal data acquisition into a highly efficient light field imaging approach: by distributing the designed or existing nonlocal image acquisition process into different views and modelling the correlations inherent in 4D light fields, CLIP can recover the 4D light field or directly retrieve refocused images from a measurement dataset even smaller than a single sub-aperture image. Under the CLIP framework, sensors of arbitrary formats—a single pixel, a linear array, or a sparse 2D area detector—can be employed for efficient light field imaging by transforming the imaging models of a single pixel camera, x-ray computed tomography and a diffuser camera⁶, to name a few. Additionally, CLIP is natively applicable to camera array systems, and promotes robustness against defective sensor measurements and severe scene occlusions. With CLIP, we seamlessly synergized multi-view with time-of-flight techniques, and demonstrated single-shot 3D imaging of texture-less scenes in an extended depth range ...”.

2. It would be good to highlight the difference between the proposed reconstruction model and prior compressive light field approaches. Both propose a linear optimization method, and include a perspective transform (i.e. shearing) to do the reconstruction so the methods seem very similar. Is the main difference between the papers the proposed capture hardware?

Response:

We thank the reviewer for the suggestion to clarify the difference between the proposed CLIP and previous compressive light field imaging methods. CLIP differs in both imaging models and implementation hardware. More fundamentally, it is a **systematic method to design and transform any imaging models that employs nonlocal image acquisition into an efficient light field imaging method.**

The perspective transform (and resultant imaging model) in compressive light field photography by Marwah et. al.³ is applied on the encoding mask, **and is static** once the mask is fixed inside the camera, while the perspective transform in CLIP is applied on the sub-aperture images and need be **numerically adjusted** to change the reconstruction focus, similar to conventional light field cameras when refocusing onto different depths. Secondly, we showed in the revised Supplementary Note 5 of Supplementary Materials that existing compressive light field imaging methods are ill-suited to 0D, 1D, or a sparse 2D detector while CLIP can accommodate detectors of arbitrary formats. Moreover, we demonstrate experimentally in the revised manuscript that CLIP can recover a 4D light field or directly reconstruct a refocused image from the same measurement data, and showed in Supplementary Note 8 that the later approach has the advantage of coping with complex scenes better.

A detailed comparison against existing methods was added in the dedicated Supplementary Note 5 of Supplementary Materials, and the 4D light field reconstruction versus direct reconstruction of refocused images are added in Supplementary Note 8, both are appended below for clarification:

Supplementary Note 5. Comparison of CLIP with compressive light field photography

Existing compressive light field imaging methods are not necessarily convolutional and can recover a 4D light field ($n_a \times n_a \times N \times N$) from a 2D image ($N \times N$). We compare them with CLIP and explain the unique advantages of CLIP in using sensors of arbitrary formats for efficient light field imaging. Most compressive light field photography methods share the roots with coded aperture imaging in using a mask (transmissive or reflective) to divide the system aperture into small patches, each modulating a sub-aperture image. The resultant sensor measurement is a weighted integration of all the sub-aperture images:

$$y_1 = \sum_{k=1}^{n_a^2} w_{1k} P_k = [w_{11}\mathbf{I}, w_{12}\mathbf{I}, \dots, w_{1n_a^2}\mathbf{I}] \begin{bmatrix} P_1 \\ P_2 \\ \vdots \\ P_{n_a^2} \end{bmatrix} \quad (17)$$

where $y_1 \in \mathbb{R}^{n^2 \times 1}$ is the vectorized sensor image, $\mathbf{I} \in \mathbb{R}^{n^2 \times n^2}$ is the identity matrix. $w_{1k} \neq w_{1j}$, and it is a scalar representing the mask transmission coefficient for the k -th sub-aperture image. $P_k \in \mathbb{R}^{n^2 \times 1}$ is the corresponding vectorized sub-aperture image. It is noted that imaging without the coding mask is equivalent to setting all the weights w_{1k} to 1. While n_a^2 different set of mask coefficients w_{jk} (and sensor measurements y_j) are typically needed to recover the light field (P_1 to $P_{n_a^2}$), Ashok⁷ and Babacan⁸ proposed to use a smaller number $m < n_a^2$ of mask

coefficients and relied on the sparsity prior for a compressive reconstruction of a 4D light field. Ashok et.al., further showed that one can use a similar coding scheme for each microlens in an unfocused light field camera, and recover the spatial image on the microlens with a sub-Nyquist measurement dataset, thereby addressing the angular-spatial resolution tradeoff in unfocused light field cameras. Nevertheless, multiple measurements are still needed in Ashok and Babacan’s methods for recovering a light field.

Marwah³ et.al., generalized the mask position to anywhere between the aperture and the sensor. When the mask is positioned close to the sensor, different sub-aperture images are modulated with sheared (and thus incoherent) mask codes before being integrated by the sensor:

$$y = \sum_{k=1}^{n_a^2} \mathbf{C}_k P_k = [\mathbf{C}_1, \mathbf{C}_2, \dots, \mathbf{C}_{n_a^2}] \begin{bmatrix} P_1 \\ P_2 \\ \vdots \\ P_{n_a^2} \end{bmatrix} \quad (18)$$

where $\mathbf{C}_k \in \mathbb{R}^{n^2 \times n^2}$ is the block diagonal matrix containing the sheared mask code. One key improvement of Marwah’s work lies in the modulation of each sub-aperture image P_k with a random code \mathbf{C}_k rather than $w_{jk} \mathbf{I}$ as in Supplementary Eq. 17, thereby improving the conditioning of the inverse problem as \mathbf{C}_k is incoherent with respect to each other. Coupled with a dictionary learning process that better sparsifies a 4D light field, Marwah’s approach can recover a full 4D light field from a single measurement, eliminating the need of changing the mask codes.

The diffuser-camera-based light field imaging^{4,5} differs from the above approaches in being convolutional: each sub-aperture image is convolved with a random nonlocal point-spread-function (PSF) before integration:

$$y = \sum_{k=1}^n \mathbf{M}_k P_k = [\mathbf{M}_1, \mathbf{M}_2, \dots, \mathbf{M}_{n_a^2}] \begin{bmatrix} P_1 \\ P_2 \\ \vdots \\ P_{n_a^2} \end{bmatrix} \quad (19)$$

with $\mathbf{M}_k \in \mathbb{R}^{n^2 \times n^2}$ being the Toeplitz convolution matrix for the random PSF in the k -th angular view. Light field imaging based on a diffuser camera can be implemented with both lens⁸ and lensless manners⁷. When being used with a lens, the PSF for each sub-aperture image is more compactly supported, leading to an efficient utilization of the sensor pixels owing to smaller boarder effects). In contrast, the lensless approach features system simplicity, and it is free from lens-aberrations.

It is now clear that the differentiating factor among existing compressive light field imaging methods is the matrix operating on each sub-aperture image. The matrices $(\mathbf{I}, \mathbf{C}_k)$ in Ashok, Babacan, and Marwah et.al. are all diagonal. As a result, the sensor resolution directly determines the spatial resolution of the recovered light field (both y and P_k are in $\mathbb{R}^{n^2 \times 1}$), making these methods ill-suited for 0D, 1D, and sparse 2D sensors. In contrast, the Toeplitz matrix \mathbf{M}_k in diffuser-camera-based light field imaging is non-diagonal, and its row vectors multiplex multiple elements of P_k into one measurement in y (owing to a nonlocal PSF). Though not being demonstrated yet, this allows in theory the recovery of a 4D light field from a sub-Nyquist measurement dataset (that is $y \in \mathbb{R}^{m \times 1}$ with $m < n^2$ while $P_k \in \mathbb{R}^{n^2 \times 1}$).

In contrast, CLIP is a systematic method for designing and transforming any imaging methods with nonlocal data acquisition into a highly efficient light field imaging approach. For a given imaging model with measurement matrix \mathbf{A} , the transformation of CLIP is achieved by splitting the measurements into different angular views, as illustrated below:

$$y = \mathbf{A}x = \begin{bmatrix} \mathbf{a}_1^T \\ \mathbf{a}_2^T \\ \vdots \\ \mathbf{a}_l^T \end{bmatrix} x \xrightarrow[\text{CLIP Step 1}]{\text{Transforming: measurement splitting}} y = \begin{bmatrix} \text{view}_1 \begin{bmatrix} \mathbf{a}_1^T \\ \vdots \\ \mathbf{a}_q^T \end{bmatrix} & \dots & \mathbf{0} \\ \vdots & \text{view}_k \begin{bmatrix} \mathbf{a}_{kq+1}^T \\ \vdots \\ \mathbf{a}_{kq+q}^T \end{bmatrix} & \vdots \\ \mathbf{0} & \ddots & \mathbf{0} \\ \mathbf{0} & \dots & \text{view}_l \begin{bmatrix} \mathbf{a}_{lq+1}^T \\ \vdots \\ \mathbf{a}_{lq+q}^T \end{bmatrix} \end{bmatrix} \begin{bmatrix} \mathbf{P}_1 \\ \mathbf{P}_2 \\ \vdots \\ \mathbf{P}_l \end{bmatrix} = \begin{bmatrix} \mathbf{A}_1 & \dots & \mathbf{0} \\ \vdots & \mathbf{A}_2 & \vdots \\ \mathbf{0} & \ddots & \mathbf{0} \\ \mathbf{0} & \dots & \mathbf{A}_l \end{bmatrix} \begin{bmatrix} \mathbf{P}_1 \\ \mathbf{P}_2 \\ \vdots \\ \mathbf{P}_l \end{bmatrix} = \mathbf{A}' \begin{bmatrix} \mathbf{P}_1 \\ \mathbf{P}_2 \\ \vdots \\ \mathbf{P}_l \end{bmatrix}, \quad (20)$$

where \mathbf{a}_k^T is a row vector, and x (an image from a single angular view) is extended to a 4D light field (\mathbf{P}_1 to \mathbf{P}_l) with $l=n_a^2$ views (sub-apertures). While the imaging model becomes block diagonal, recovering the light field is equivalent to solve each sub-aperture image \mathbf{P}_k with a corresponding sub-measurement matrix \mathbf{A}_k . We can further exploit the correlations (redundancy) in the 4D light field by solving Supplementary Eq. 20 with appropriate sparsity based regularizations, as used in compressive light field imaging methods³⁻⁵. It is noteworthy that the elemental matrix \mathbf{A}_k is not diagonal as \mathbf{I} or \mathbf{C}_k , a key fact that enables CLIP to use 0D or 1D sensors for light field imaging. We demonstrated 4D light field recovery using CLIP in Supplementary Note 7.

The second key differentiating factor of CLIP is explicit modeling of the correlations among sub-aperture images as $\mathbf{P}_k = \mathbf{B}_k h$ via light field propagation assuming a uniform angular intensity distribution as derived in Supplementary Note 1. This simplifies Supplementary Eq. 20 to the CLIP equation 3 in the main text:

$$y = \begin{bmatrix} \mathbf{A}_1 & \dots & \mathbf{0} \\ \vdots & \mathbf{A}_2 & \vdots \\ \mathbf{0} & \ddots & \mathbf{0} \\ \mathbf{0} & \dots & \mathbf{A}_l \end{bmatrix} \begin{bmatrix} \mathbf{P}_1 \\ \mathbf{P}_2 \\ \vdots \\ \mathbf{P}_l \end{bmatrix} \xrightarrow[\text{CLIP Step 2}]{\mathbf{P}_k = \mathbf{B}_k h} y = \begin{bmatrix} \mathbf{A}_1 & \dots & \mathbf{0} \\ \vdots & \mathbf{A}_2 & \vdots \\ \mathbf{0} & \ddots & \mathbf{0} \\ \mathbf{0} & \dots & \mathbf{A}_l \end{bmatrix} \begin{bmatrix} \mathbf{B}_1 h \\ \mathbf{B}_2 h \\ \vdots \\ \mathbf{B}_l h \end{bmatrix} = \begin{bmatrix} \mathbf{A}_1 \mathbf{B}_1 \\ \mathbf{A}_2 \mathbf{B}_2 \\ \vdots \\ \mathbf{A}_l \mathbf{B}_l \end{bmatrix} h = \mathbf{A}'' h. \quad (21)$$

This step has the advantage of enabling more complicated images to be recovered without the need of finding/learning a better sparsifying basis for the 4D light field, which is an important step in Marwah's work. We show this advantage in Supplementary Note 8.

The computation complexity of compressive light field photography and CLIP depends on the light field resolution and the applied regularization method under the framework of regularization by denoising (see **Methods**). In CLIP, each iteration involves a pass of \mathbf{A}'' and

\mathbf{A}'^T along with a denoising step. The complexity for the shearing operation and matrix \mathbf{A} is $\mathcal{O}(n_a^2 N^2)$ and $\mathcal{O}(mN^2)$ respectively, leading to a total complexity of $\mathcal{O}((n_a^2 + m)N^2)$ for both \mathbf{A}'' and \mathbf{A}'^T . The complexity of BM3D and TV denoising for regularization is directly related to the image size as $\mathcal{O}(kN^2)$, with k being a denoiser-dependent constant. Therefore, the total complexity of CLIP image recovery is $\mathcal{O}((2m + 2n_a^2 + k)N^2)$ per iteration. In comparison, while the complexity for \mathbf{A}' and \mathbf{A}'^T in Supplementary Eq. 20 for retrieving the 4D light field remains $\mathcal{O}(mN^2)$ owing to the block diagonal structure, the denoising complexity of a 4D light field becomes $\mathcal{O}(kn_a^2 N^2)$, resulting in a total complexity of $\mathcal{O}((2m + kn_a^2)N^2)$. Similarly, we can analyze the computation complexity per iteration for compressive light field imaging methods based on the model in Supplementary Eq. 17 to 19. Supplementary Table 1 summarizes the characteristics of CLIP and compressive light field photography. It is worth noting that the computation complexity of Marwah’s work does not account for the dictionary learning process, and the regularization is applied on the entire light field. Also, the convolution model of the diffuser-camera is accelerated by FFT.

Supplementary Table 1 Comparison of CLIP and compressive light field photography

Methods	Sensor	Light field size	Measurement data size	Compression axis	Computation complexity	
Ashok ⁷	2D	$n_a \times n_a \times N \times N$	$r \times N \times N$	Angular or spatial	$\mathcal{O}((2r + k)n_a^2 N^2)$	
Babacan ⁸	2D	$n_a \times n_a \times N \times N$	$r \times N \times N$	Angular	$\mathcal{O}((2r + k)n_a^2 N^2)$	
Marwah ³	2D	$n_a \times n_a \times N \times N$	$N \times N$	Angular	$\mathcal{O}((2 + k)n_a^2 N^2)$	
Cai ⁷ , Antipa ⁸	2D	$n_a \times n_a \times N \times N$	$N \times N$	Angular	$\mathcal{O}((4 \log N + kn_a^2)N^2)$	
CLIP	0D, 1D, 2D	$n_a \times n_a \times N \times N$	$m (\leq N \times N)$	Angular and/or spatial	4D light field	$\mathcal{O}((2m + kn_a^2)N^2)$
					Refocus image	$\mathcal{O}((2m + 2n_a^2 + k)N^2)$

Supplementary Note 8. CLIP 4D light field reconstruction versus direct reconstruction

While CLIP can recover a 4D light field as demonstrated in previous note, we show here that directly recovering a refocused image can better accommodate complex scenes, particularly for imaging with lower dimension (1D or 0D) sensors. Marwah’s work relied on a dictionary learning process to obtain a representation basis to better sparsify the 4D light field, thereby attaining excellent 4D light field reconstruction for complex scenes. On the other hand, Antipa⁴ pointed out that improper regularization of the 4D light field in diffuser-based camera can degrade (or even destroy) the angular information in the light field.

In contrast, CLIP doesn’t rely on high quality 4D light field reconstruction to obtain excellent refocused images: CLIP’s complementary measurements among sub-apertures can significantly improve the refocused images despite the recovered 4D light field may not be of high quality, which is the case under the compressive regime. Further, CLIP can directly recover a refocused image like coded-aperture and wavefront-coding methods to accommodate complex scenes better, as explained in previous section. We demonstrate this via a synthetic study for the synthetic scene 2 and an experimentally acquired light field from the ‘letter scene’,

using a sampling ration of $SR=1$. During the reconstruction for the 4D light field, the regularization parameter is tuned from to obtain a best refocused image from the light field data. Supplementary Figure 11 shows the recovered 4D light field and refocused images for the two scenes under the CLIP-1D (a and b) and CLIP-0D (c and d) implementations, with the NMSE listed in Supplementary Table 4. It is noted that while the light field suffers from significant background signals and noises, the refocusing processing coherently assembles CLIP’s complementary imaging across the sub-apertures to yield substantially better refocused image. Moreover, CLIP’s direct reconstruction further improved the quality of the refocused image by rendering more image details and a higher contrast.

Supplementary Table 4. NMSE of 4D and direct CLIP reconstructions

		Scene	Method	s=-1.0	s=1.0				
CLIP-1D	1		4D recon.	11.98%	6.08%	CLIP-0D	4D recon.	10.47%	10.67%
			Direct recon.	3.66%	4.16%		Direct recon.	6.23%	7.35%
	2		4D recon.	18.68%	8.48 %		4D recon.	13.02%	7.53%
			Direct recon.	6.33%	6.75%		Direct recon.	4.41%	4.05%

Supplementary Figure 11. 4D light field reconstruction versus direct reconstruction of refocused images by CLIP. a-b, CLIP-1D reconstruction for the synthetic scene and the experimental ‘letter’ scene. c-d, CLIP-0D reconstruction for the two scenes. The sampling ratio of CLIP is fixed at $SR=1$.

3. The paper uses algorithm speed as a major factor to motivate the work, but doesn't do a lot to quantify or verify that statement. The authors mention runtimes of methods in some places, but often the runtimes include a bulk time for different methods like line of sight and NLOS reconstruction. There is also no comparison of computational or memory complexity. I think the paper needs to include some meaningful comparison to alternative methods and a discussion of the expected improvements in performance over prior methods. It also needs to provide complete and structured information about the actual execution speeds and put those in some meaningful context.

Response:

We thank the reviewer for this suggestion. We analysed and compared the computation complexity per iteration for CLIP reconstruction with compressive light field imaging methods in Supplementary Table 1 of the revised Supplementary Note 5, please see the excerpt in *Response to Comment 2* above for details.

Regarding NLOS imaging with CLIP camera, there are two parts for the reconstruction: CLIP reconstruction of the (x, y, t) data cube in the first part and then applying the hybrid time-frequency domain algorithm to recover a 3D hidden scene in the second part. CLIP can accelerate (x, y, t) data acquisition (down to a single shot), but the iterative reconstruction is not fast enough for real-time imaging. To solve this problem, we show in Supplementary Note 13 (originally Supp. Note 9) that a fast 'adjoint reconstruction' of CLIP can be used for NLOS imaging at the expense of a degraded imaging robustness against noises. We also compared the proposed hybrid time-frequency domain algorithm with alternative methods in terms of computation and memory complexity in the revised Methods Section of the main text, along with the execution time that includes CLIP reconstructions of the (x, y, t) data cube. The comparisons are excerpted below to clarify these points.

“...with both the curved and virtual plane being sampled with a spatial resolution of N^2 , time-domain migration has a computational complexity of $o(N^4)$ instead of $o(N^5 \log N)$... Combined with the complexity of $o(N^3 \log N)$ for the frequency-domain phasor field reconstruction, the total complexity for the hybrid time-frequency domain reconstruction is $o(N^4)$, still orders of magnitude faster than time domain methods. The memory complexity for the frequency-domain phasor field and relevant f - k migration reconstruction have been analyzed in the literature to be $o(N^3)$ and $\sim o(50N^3)$ ⁹, respectively. The time-domain migration in Eq. (9) has a memory complexity of $o(N^3)$ in order to store the propagated signals $f(r_v, t)$ on a virtual plane, leading to a total complexity of $o(N^3)$ for the hybrid time-frequency domain reconstruction method...

... For a $128 \times 128 \times 128$ imaging volume with a spatiotemporal data cube of $125 \times 125 \times 1016$, the NLOS reconstruction time is ~ 0.03 seconds, which can reach a 30 Hz video rate. The actual bottleneck lies in the iterative CLIP reconstruction of the spatiotemporal data cube on the wall,

which takes about 2.0 seconds. However, we show in Supplementary Note 9 that a fast CLIP solution via the adjoint operator can reduce the reconstruction time to 0.01 seconds for NLOS imaging at the expense of noise robustness. Table 1 summarizes the computation and memory complexity of the hybrid time-frequency domain reconstruction method against the time-domain phasor field method and f - k migration for NLOS imaging with curved surfaces. It is noteworthy that the complexity of f - k migration includes the necessary preprocessing step for coping with curved surfaces, and its execution time is obtained by CPU processing with a downsampled spatiotemporal data cube of $(32 \times 32 \times 512)$ instead of $(125 \times 125 \times 1016)$. Conforming with the complexity analysis, the preprocessing step is more time consuming than the actual reconstruction in f - k migration.

Table 1 Comparison of NLOS imaging computation and memory complexity

Algorithms	Computational complexity	Memory complexity	Execution time		Total reconstruction (seconds)
			CLIP reconstruction (seconds)	NLOS reconstruction (seconds)	
f - k migration	$o(N^5 \log N) + o(N^3 \log N)$	$\sim o(50N^3)$	0	15.1 + 0.65 (CPU)	15.7 (CPU)
Time-domain phasor field	$o(N^5)$	$o(N^3)$	0	0.40 (GPU)	0.40 (GPU)
Hybrid time-frequency method	$o(N^4)$	$o(N^3)$	2.0 or 0.01 (adjoint method)	0.03 (GPU)	2.03 or 0.04 (GPU)

4. Similarly, the paper talks about the robustness of the work to missing or erroneous pixels, but does not actually do anything to test that.

Response:

We thank the reviewer for raising this point. The robustness against missing pixels was demonstrated to some extent by using a sparse 2D detector for light field imaging in the original Supplementary Figure 4 of Supplementary Materials. We added both experimental and synthetic results in dedicated Supplementary Note 9 of the revised Supplementary Materials to further demonstrate and quantify its robustness against to erroneous or missing pixels, which is referred to in the revised main text as “... endows CLIP with imaging robustness against defective pixels or scene occlusions. Because the complete scene is encoded in any subset of the measurements, image recovery is not substantially affected by a fraction of defective pixel readings, despite that the conditioning of image reconstruction might deteriorate (Supplementary Note 9)...”

Supplementary Note 9 is appended below to clarify this point.

“ **Supplementary Note 9. CLIP robustness** ”

The robustness against missing pixels of CLIP is demonstrated to some extent by imaging with sparse 2D detectors in Supplementary Note 5. Here, we further test the robustness of CLIP against erroneous measurements. Two typical errors are dead (or missing) pixels and saturated sensor readings. We tested the case that the measurement containing both types of errors by

first normalizing the measurement data, and then randomly setting part of the measurement to 0 (dead) or 1 (saturated). The error induced by defective measurement is evaluated by NMSE for both the raw measurement data and reconstructed images. Fixing the sampling ratio SR at 1, we varied the percentage of the erroneous measurement from 0.1% to 1% for the experimental data in CLIP-0D, and 1% to 10% for the synthetic data in CLIP-1D. Supplementary Figure 12 shows the CLIP imaging results and the corresponding NMSEs are summarized in Supplementary Table 5. Owing the nonlocal data acquisition strategy and the regularization step, the reconstructed image error in CLIP is substantially smaller than the error in the raw measurements, making it more robust than classic imaging methods.

Supplementary Table 5. NMSE of CLIP reconstruction with erroneous measurement

CLIP-0D				CLIP-1D					
Error percent		0.1%	0.2%	1%	Error percent		1%	5%	10%
Exp. Scene 1	Data error	5.68%	9.9%	37.2%	Syn. Scene 1	Data error	14.53%	47.90%	66.39%
	Image error	1.54%	2.08%	11.24%		Image error	5.07%	12.1%	15.2%
Exp. Scene 2	Data error	22.52%	36.29%	75.88%	Syn. Scene 2	Data error	8.99%	34.83%	53.43%
	Image error	4.50%	4.7%	6.5%		Image error	1.68%	6.05%	9.38%

Supplementary Figure 12. CLIP reconstruction with different amount of erroneous measurement data. The error in the title is listed as percentage of the total measurement number.

Syn.: synthetic; Exp.: experimental.

...

5. I'm a little confused about the actual setup used in the different demonstrations. The imaging setup with the streak camera and lenslet array should result in a light field array with a diameter similar to the slit of the streak camera. So a centimeter or two. An array of that size should not be big enough to image around the occlusions they create. The authors explain the geometry that determines the permissible size of the occluder in the supplement. I think some added clarification and maybe a sketch of the setup is needed here.

Response:

In principle, it is the relative scale between the camera baseline and object scene that matters for imaging through occlusions. We added a photograph for the setup of the proof-of-concept experiments in Supplementary Note 13b, and gave numerical details in the revised Supplementary Note 10 to clarify this point, which reads “...A photograph of system setup for the dynamic imaging experiment is shown in the bottom of Supplementary Figure 13b, where the camera baseline L is ~ 15 mm, and the occluder was placed at approximately $d=50$ mm (or ~ 40 mm in the static studies) from the lenslet array. For an occluder with width $D \approx 6$ (or 10) mm, the inactive region is hence $l_d \approx 33$ (or 80)mm. The object was positioned at a distance ~ 70 mm (or >90 mm for different static studies) from the occluder to avoid falling into the inactive region...”

Supplementary Figure 13. 3D imaging through occlusions. **b** Geometry for calculating the inactive region caused by the occluder, and the experimental setup for dynamic imaging studies.

6. NLOS imaging using non-planar and changing relay walls has been demonstrated by La Manna et. al. (43). The method presented here is probably faster, but a little more discussion of the methods is probably necessary. The

paper about the FK migration algorithm that the authors use also describes reconstruction from non-planar surfaces.

Response:

We agree that NLOS imaging with dynamic non-planar relay wall has been demonstrated by La Manna et. al., where they scanned a collimated laser beam rather than the SPAD detector for 2D recording of the time-of-flight data. The reception point was fixed at a stationary point, sidestepping the depth-of-field problem of the detection optics. When using array detectors to accelerate NLOS imaging acquisition on curved surfaces, Manna's method will suffer the depth-of-field problem as usual. In contrast, CLIP can use a 1D array detector for fast imaging with curved surfaces.

The f - k migration algorithm indeed can be adapted for NLOS imaging with curved surfaces, but its confocal imaging process still suffers from a long acquisition time, and as compared in Table 1 of the revised Methods Section (see *Response to Comment 3*), its preprocessing step to cope with non-planar surfaces is actually more time-consuming than the actual f - k migration step or the time-domain phasor field method.

Regarding this, we added a discussion on these two points in the section of "NLOS imaging with curved and disconnected surfaces" that reads "... The ToF-CLIP camera addresses this critical need for real-time mapping of the relay surface via built-in flash LiDAR imaging. More importantly, it can accommodate a non-planar surface geometry for NLOS imaging **using array detectors** with its light field capability. **Paired with a proposed hybrid time-frequency domain reconstruction algorithm, which can handle general surfaces with a computational complexity of $o(N^4)$ (Methods), ToF-CLIP can attain real-time NLOS imaging with arbitrary curved surfaces. While NLOS imaging with a dynamic and curved surface has been demonstrated by Manna¹⁰ et. al., its reception point was fixed at a stationary point rather than being on the dynamic surface, making it inapplicable for real-time imaging with array detectors. Similarly, the preprocessing step¹¹ proposed by Lindell et.al. that adapts the f - k migration reconstruction algorithm to deal with slightly curved surfaces in confocal NLOS imaging has a computational complexity of $o(N^5 \log N)$, which is higher than the time-domain phasor field method and thus inefficient for real-time reconstruction...**"

7. The computationally most challenging step of FK migration is the necessary interpolation in the Fourier domain. That is what drives complexity and memory use and tends to create artifacts. The published code the authors use addresses this problem by oversampling the reconstruction in the Fourier domain resulting in very high memory use. In that light, the statement, that FK does not require interpolation is misleading. The authors refer to interpolation in the time domain. FK instead needs interpolation in the Fourier domain.

Response:

We agree with the reviewer that FK migration need to interpolate from a spherical coordinate onto a Cartesian one in the Fourier domain that causes high memory usage.

We clarify that the frequency-domain NLOS reconstruction method that we used was the fast frequency-domain phasor-field method proposed by Liu et. al., which consumes much less memory.

The interpolation in time domain is to correct for the perspective distortion of the recording camera on a non-planar surfaces, thereby yielding a regular 2D grid sampling pattern on the virtual plane to facilitate subsequent frequency-domain NLOS reconstruction (otherwise, a nonuniform FFT based NLOS reconstruction algorithm needs to be developed). This interpolation is not needed in the proposed hybrid time-frequency domain method because the waves can be directly migrated in the time-domain to a regular 2D grid on the virtual plane.

Regarding this, we clarified in the revised Methods section that the second part of the hybrid time-frequency domain reconstruction is the frequency-domain phasor-field that reads “ ... The hybrid frequency-time domain reconstruction method proposed here first converts the spatiotemporal measurement on a curved surface $y_r(\mathbf{r}_p, t)$ onto a virtual plane via wave propagation in time domain and then reconstruct the hidden scenes with existing efficient frequency-domain **phasor field method**⁹...”

Reviewer 2

The manuscript reports on a method for light field photography in which, in essence, instead of acquiring L different images from different view points, only one pixel or one line of each image is acquired but each one from a different perspective. These are then combined together through an minimisation approach that relies on a "shear" operator that models how the various parts of the scene are captured at varying view points and then registered in a single final image.

The idea is clever and seems to deliver very promising results. The authors show many different possible implementations of the technique, ranging from 2D imaging, flash lidar to non line of sight imaging.

I am not personally convinced that the NLOS imaging results are that significant compared to the state of the art. However, the other results do look convincing, including the measurements in the presence of occluders. The video material provided is also very convincing.

The work is very carefully prepared with sufficient details to reproduce the results. My only comment is that the supplementary information is actually very much integral to the main work as many or most of the actual results are presented there. This is probably a choice based on the fact that the authors present so many different cases that it is hard to show all results in the main

text. But this is just a stylistic choice and does not impact the importance of the work itself.

I therefore suggest acceptance of this work for publication without any need for revision.

Response:

We appreciate the reviewer's positive comments on our work. Regarding the NLOS imaging methods, the quality is not yet state-of-art because it is imaged with a small number of time-of-flight sensors (a 1D sensor in our demonstration) in a **snapshot, scanless manner** (<0.1 s), which causes a high compression factor (~20) for NLOS imaging.

Reviewer 3

In this manuscript, the authors report their development of an imaging method which they call "compact light field photography (CLIP)". They claim that CLIP enables three-dimensional imaging with fewer detectors compared to conventional light field photography methods. They demonstrated volumetric imaging by combining CLIP with other imaging techniques, such as time-of-flight, LiDAR, and non-line-of-sight 3D imaging. The main argument of this work is that the data size can be reduced compared to conventional light field photography, which is advantageous for large-scale, high-dimensional photography. However, I do not think the quality of the manuscript meets the publication criteria of Nature Communications in terms of novelty, quality of presentation, and impact of the results. Detailed comments are listed below:

Response:

We appreciate the reviewer's extensive and constructive comments on our work. Extensive revisions have been made on the manuscript and supplementary materials accordingly to address the raised points, as detailed below.

1. I doubt the effectiveness of their approach toward the realization of dynamic 3D imaging. The authors perform data compression optically by using specially arranged sensors (a single pixel, a linear array, or a sparse 2D area detector). Whether this compression works for retrieval of 3D images depends on the scene (as long as the restricted isometric property of the measurement matrix is not evaluated). The authors' approach seems to inherently lack generality.

Response:

We thank the reviewer for raising the important point on evaluating the RIP (restricted isometric property) of the measurement matrix when working in the compressive

regime. We clarified that, while a major appeal of CLIP is to use a limited sensor budget to acquire large-scale light fields, **it is not necessarily confined to the compressive regime for directly solving a refocused image**. Also, CLIP can well accommodate, but is not limited to, those special sensor formats. When working in the compressive regime, we show in the revised manuscript that CLIP is general enough for recovering *structured-sparse* signals such as natural images.

As detailed in the Revised Supplementary Note 5 (Please see *Response to Comment 3 of Reviewer 1*) that articulates the difference between CLIP and compressive light field imaging methods, CLIP can transform any imaging model $y = \mathbf{A}x$ ($\mathbf{A} \in \mathbb{R}^{m \times n}$) that employs nonlocal data acquisition into a light field imaging method. The resultant CLIP equation $y = \mathbf{A}'x$ has the same dimension with measurement matrix \mathbf{A} (that is, $\mathbf{A}' \in \mathbb{R}^{m \times n}$). As a result, it is not necessarily under-determined and works equally well for imaging methods using dense 2D sensor arrays—it is proved in Supplementary Note 3 that CLIP can include coded-aperture and wavefront-coding based light field imaging methods as special cases. The motivation of using 0D and 1D sensor is that they are far more accessible for imaging at the ultrafast time scale or in the infrared/Terahertz spectral band, for which existing compressive light field imaging methods are ill-suited, as proved in Supplementary Note 3 and 5. Also, the mathematical model of CLIP with 0D and 1D sensors are transformed from the imaging model of the single pixel camera and x-ray computed tomography respectively, which have been demonstrated to show the generality for imaging applications in practice when working in the compressive regime.

Mathematically, the generality of CLIP in the compressive regime can be evaluated by computing the RIP constant of the measurement matrix \mathbf{A}' as mentioned by the reviewer. However, RIP is only a sufficient condition, and evaluating RIP is a NP-hard problem. We added extensive numerical tests to demonstrate the generality of CLIP under the *structured-sparse* signal model in the dedicated Supplementary Note 6 of Supplementary Materials, and stressed in the revised Methods section that CLIP has the generality to recover *structured-sparse* signals such as natural images, which reads “ ... It is worth noting that while recovering the 4D light field is always compressive in CLIP, directly retrieving a refocused image is not necessarily the same. Still, a major appeal of CLIP is to use a small number of sensors for recording a large-scale light field, which typically falls into the compressive sampling regime. In this case, we show in Supplementary Note 6 that while the imaging model designed in or transformed by CLIP may not satisfy the restricted isometry property (RIP) to guarantee uniform recovery of arbitrary images in the classic *sparse* signal model, CLIP has the generality in the *structured-sparse* signal model and hence remains applicable for practical imaging applications.”

The added Supplementary Note 6 is appended below for a detailed explanation.

“Supplementary Note 6. Generality of CLIP

While recovering a 4D light field is always under-determined in CLIP and compressive light field photography methods, directly recovering a refocused image by CLIP is not necessarily

the same. As a result, CLIP isn't bounded to the compressive regime, though one of its major appeal is to record a large-scale light field with a highly limited sensor budget. When working in the compressive regime, it is important to evaluate whether the system matrix \mathbf{A}' of CLIP supports a uniform recovery of arbitrary s -sparse vectors (vectors with at most s non-zero entries) in the classic sparse signal model by computing the restricted isometry property (RIP) of matrix \mathbf{A}' . However, RIP is not a necessary condition and computing the RIP constant is an NP-hard problem. Up to now, only a limited types of matrices have been proven to satisfy RIP with an exponentially high probability. On the other hand, it was shown¹² that there is an absence of RIP in a range of practical compressive imaging applications, and yet, experimental image recovery is excellent. These applications include compressive x -ray tomography, MRI, and single pixel cameras. The work of Bastounis¹² and Roman¹³, among other similar works¹⁴, attributed the correct recovery of image x to the *structured-sparsity* of x (that is, the sparsity of x has a structure instead of exhibiting an arbitrary pattern), and together with an extended concept of *RIP in levels*, explained the success of these compressive imaging methods in practice, despite that their measurement matrices failed to satisfy the classic RIP. As natural images are highly structured, and CLIP with 0D and 1D sensors are transformed from the single pixel cameras and x -ray tomography methods respectively, it is expected that CLIP can attain similar imaging performance in practice.

We followed the philosophy of generalized flip test proposed by Roman et.al.¹³ to evaluate the general applicability of CLIP under the *structured-sparsity* signal model. This idea of the test is to evaluate the reconstruction quality of different images with the same sparsity. To generate such images, we applied shift, flip, rotation operation on some image part, and evaluated the reconstruction error using normalized mean square errors (NMSE). As CLIP deals with light field data, these operations should be applied to 3D objects. To this end, the 3D scenes were modelled in Blender software for rendering the 4D light field data on a regular 2D grid.

Throughout the manuscript, synthetic CLIP measurement with 1D and 0D sensors were obtained as follows. In CLIP-0D, each sub-aperture image is encoded with random binary codes to yield $m_k=m/l$ single-pixel readings. For CLIP-1D, the measurements are obtained in three steps: a) generate m/N projection angles α uniformly in the range of $[0, 180^\circ]$; b) randomly permute the angles α and distribute evenly into the l sub-apertures; c) calculate for each sub-aperture image the projection data along the assigned angles. The sampling ratio (SR) is defined as the quotient between the total number of measurements m and the image size N^2 (rather than the 4D light field). For this test, we fixed SR at 0.5.

Supplementary Figure 6 and 7 show the CLIP imaging results for two different scenes under various focus settings, with NMSE listed on Supplementary Table 2. It is noted that CLIP consistently yields a NMSE below 10% for SR=0.5, indicating its generality in coping with natural scenes when working in the compressive regime. Further results demonstrating the generality of CLIP are given in Supplementary Note 11, which employs CLIP (with different sampling ratio SRs) to represent experimentally acquired light field data for scenes with different BRDFs.

Supplementary Table 2. NMSE of CLIP reconstruction for synthetic scene 1 and 2

	s	CLIP-1D				CLIP-0D			
		Original	Shift	Flip	Rotate	Original	Shift	Flip	Rotate
Scene 1	-0.8	7.28%	8.75%	8.13%	7.26%	8.75%	9.66%	9.70%	9.36%
	-0.1	9.53%	7.10%	7.14%	7.59%	8.71%	7.97%	9.22%	8.18%
	0.5	7.22%	7.96%	7.17%	6.38%	8.97%	9.34%	8.52%	8.39%

Scene 2	-0.6	4.71%	4.34%	4.88%	4.54%	5.40%	5.37%	5.20%	4.64%
	0	5.52%	3.70%	3.67%	4.28%	5.47%	5.57%	5.54%	5.54%
	0.8	6.48%	4.21%	4.40%	4.89%	5.22%	4.86%	4.59%	5.08%

Supplementary Figure 6. Generalized flip test of CLIP reconstruction for synthetic scene 1 with $SR = 0.5$. The ground truth light field size is $8 \times 8 \times 128 \times 128$, and the measurement data size is 64×128 , leading to a data reduction of 128.

Supplementary Figure 7. Generalized flip test of CLIP reconstruction for synthetic scene 2 with $SR = 0.5$. The ground truth light field size is $8 \times 8 \times 128 \times 128$, and the measurement data size is 64×128 .

2. Also, optical compression accompanies the loss of data due to difficult-to-control factors. For example, in the authors' setup in Fig. S5d, the angles and positions of cylindrical lenses, the widths of slits, and the distance between the lenses and the sensor critically change the intensity profile on the sensor. In addition, aberration due to the imperfection of the lenses induces loss of information. As long as one can acquire the entire image, it should be taken. If compression is needed, we can do it by post processing using an FPGA in a high-throughput, lossless, and reproducible manner.

Response:

We agree with the reviewer that when a suitable 2D sensor is available for a target application, acquiring the image at/over the Nyquist rate and then compressing it in post processing will be advantageous in terms of fidelity. However, this is not always feasible, and the motivation for compressive sampling is mainly two-folds.

First and foremost is the availability (and economy) of suitable detectors for acquiring the signal of interest. Currently, there is no ultrafast detectors similar to consumer-grade 2D CMOS or CCD image sensors for snapshot acquisition of large-scale time-of-flight data or any similarly high-dimensional data such as hyperspectral images. Existing ultrafast cameras are in the format of a single pixel (SPAD, PMT etc.), a linear array (streak camera or linear array SPAD), or a sparse 2D array (state-of-the-art SPAD array has a relatively low fill factor below 50%). As a result, a slow scanning (spatial and/or temporal) mechanism is needed for 2D time-of-flight (or hyperspectral) imaging. Other applications for which 0D and 1D sensors are more accessible include imaging in the infrared and Terahertz region, where the detector resolution remains low. It is the limited sensor budget that hampers light field imaging in these applications.

The second motivation is the compressibility of natural signals, especially high-dimensional signals. As pointed out by the reviewer, signal compression is typically done by post-processing after a full acquisition. However, for applications suffering from detector availability issues but dealing with highly compressible signals, compression in sampling phase can become advantageous because much fewer sensor measurements will be needed. Indeed, the compressibility of natural images has been well-exploited in many imaging applications. For example, x-ray CT and MRI imaging has adopted compressive sampling to reduce the radiation dose and imaging time. The compressibility of 4D light fields or natural images is also a key ingredient in existing compressive light field imaging methods.

The factors affecting optical compression is accounted for by a system calibration step, as typically done in other computational imaging methods. In theory, the nonlocal sampling and *structured-sparse* signal recovery strategy of CLIP could potentially be more robust against information loss and lens aberration of the optical system. The 0D implementation does not need a lens as demonstrated in the single pixel cameras, and CLIP with a 1D sensor tends to suffer from less aberration than conventional imaging because there is no optical power in the invariant axis of the cylindrical lens. The setup in Fig. S5d differs from a coded aperture camera only in replacing the spherical lens-system with a cylindrical one that modifies the ideal point spread function from a point into an angled line. The effects of other factors on signal intensity on the sensor — the positions of cylindrical lenses, the widths of slits, and the distance between the lenses and the sensor, remain the same as that in coded aperture cameras. For example, a change in the distance of the lens and sensor will defocus the image signal (with a circular blur bokeh being replaced by an elliptical one). The width of slits defines the encoding resolution. The lens position relative to the sensor

determines the imaging field of view. All these factors (except aberrations) are obtained after system alignment and calibration.

Regarding this, we clarified the calibration step for Fig. S5d in revised Supplementary Note 5, which reads “ ... It is noted that the implementation for randomly coded line-shape PSF is very similar to the coded-aperture camera, with the camera lens and image sensor being replaced by a cylindrical one and 1D sensor respectively. Like coded-aperture imaging therefore, a one-time calibration step for the camera will be needed to retrieve PSF on the sensor by imaging a point source and scanning the 1D sensor along the other dimension ...”

Also, we included a dedicated Supplementary Note 9 on evaluating the robustness of CLIP under information loss (in the form of missing and erroneous measurements). Please see *Response to Comment 4 of Reviewer 1* for more details.

3. The definition of CLIP is unclear. The authors' statement "To address these challenges, we present compact light field photography (CLIP) to sample dense light fields with a drastically improved efficiency and flexibility. By employing nonlocal image acquisitions and distributing a complete acquisition process into different views, CLIP enables light field imaging with a measurement dataset smaller than a single sub-aperture image and remains natively applicable to camera array systems" sounds no more than compressed light field photography, which has been thoroughly investigated.

Response:

Because both Comment 3 and 4 concerned the distinction between CLIP and existing compressive light field photography methods, we address them together in our *Response to Comment 4*.

4. Even with the authors' explanation, "Unlike previous compressive light field cameras^{23–25} that decode a densely sampled 2D image into a full 4D light field, CLIP features the unique capability of utilizing a small number of sensors arranged in arbitrary formats—a single pixel, a linear array, or a sparse 2D area detector—for light field imaging", the difference is unclear because one can easily reduce the effective number of data points on a CCD /CMOS camera by pixel binning.

Response:

We agree that pixel binning or extraction can reduce the measurement easily and effectively, but this will equally reduce the imaging (or light field) resolution, and require the intended applications to have a dense 2D CCD/CMOS camera to begin with. In contrast, CLIP can use a sensor of a limited resolution, such as 0D or 1D sensors, for

efficient light field imaging by transforming an appropriate imaging model that employs nonlocal data acquisition. As examples, CLIP transformed the imaging model of the single pixel camera and x-ray CT for efficient light field imaging with a single pixel and a 1D sensor respectively in the manuscript.

Moreover, we demonstrated experimentally in the revised manuscript that CLIP can recover a 4D light field or directly reconstruct a refocused image from the same measurement data (see *Response to Comment 5* below for details). While existing compressive light field imaging methods recover a 4D light field from a densely sampled 2D image, we showed in Supplementary Note 8 that the CLIP's approach of directly reconstructing a refocused image has the advantage of coping with complex scenes better. A detailed comparison against existing compressive light field imaging methods was added in the dedicated Supplementary Note 5 of Supplementary Materials, and the direct reconstruction of refocused images is compared with 4D light field reconstruction in Supplementary Note 8, both are appended below for clarification.

Supplementary Note 5. Comparison of CLIP with compressive light field photography

Existing compressive light field imaging methods are not necessarily convolutional and can recover a 4D light field ($n_a \times n_a \times N \times N$) from a 2D image ($N \times N$). We compare them with CLIP and explain the unique advantages of CLIP in using sensors of arbitrary formats for efficient light field imaging. Most compressive light field photography methods share the roots with coded aperture imaging in using a mask (transmissive or reflective) to divide the system aperture into small patches, each modulating a sub-aperture image. The resultant sensor measurement is a weighted integration of all the sub-aperture images:

$$y_1 = \sum_{k=1}^{n_a^2} w_{1k} P_k = [w_{11}\mathbf{I}, w_{12}\mathbf{I}, \dots, w_{1n_a^2}\mathbf{I}] \begin{bmatrix} P_1 \\ P_2 \\ \vdots \\ P_{n_a^2} \end{bmatrix} \quad (17)$$

where $y_1 \in \mathbb{R}^{n^2 \times 1}$ is the vectorized sensor image, $\mathbf{I} \in \mathbb{R}^{n^2 \times n^2}$ is the identity matrix. $w_{1k} \neq w_{1j}$, and it is a scalar representing the mask transmission coefficient for the k -th sub-aperture image. $P_k \in \mathbb{R}^{n^2 \times 1}$ is the corresponding vectorized sub-aperture image. It is noted that imaging without the coding mask is equivalent to setting all the weights w_{1k} to 1. While n_a^2 different set of mask coefficients w_{jk} (and sensor measurements y_j) are typically needed to recover the light field (P_1 to $P_{n_a^2}$), Ashok⁷ and Babacan⁸ proposed to use a smaller number $m < n_a^2$ of mask coefficients and relied on the sparsity prior for a compressive reconstruction of a 4D light field. Ashok et al., further showed that one can use a similar coding scheme for each microlens in an unfocused light field camera, and recover the spatial image on the microlens with a sub-Nyquist measurement dataset, thereby addressing the angular-spatial resolution tradeoff in unfocused light field cameras. Nevertheless, multiple measurements are still needed in Ashok and Babacan's methods for recovering a light field.

Marwah³ et al., generalized the mask position to anywhere between the aperture and the sensor. When the mask is positioned close to the sensor, different sub-aperture images are modulated with sheared (and thus incoherent) mask codes before being integrated by the sensor:

$$y = \sum_{k=1}^{n_a^2} \mathbf{C}_k P_k = [\mathbf{C}_1, \mathbf{C}_2, \dots, \mathbf{C}_{n_a^2}] \begin{bmatrix} P_1 \\ P_2 \\ \vdots \\ P_{n_a^2} \end{bmatrix} \quad (18)$$

where $\mathbf{C}_k \in \mathbb{R}^{n^2 \times n^2}$ is the block diagonal matrix containing the sheared mask code. One key improvement of Marwah's work lies in the modulation of each sub-aperture image P_k with a random code \mathbf{C}_k rather than $w_{jk}\mathbf{I}$ as in Supplementary Eq. 17, thereby improving the conditioning of the inverse problem as \mathbf{C}_k is incoherent with respect to each other. Coupled with a dictionary learning process that better sparsifies a 4D light field, Marwah's approach can recover a full 4D light field from a single measurement, eliminating the need of changing the mask codes.

The diffuser-camera-based light field imaging^{4,5} differs from the above approaches in being convolutional: each sub-aperture image is convolved with a random nonlocal point-spread-function (PSF) before integration:

$$y = \sum_{k=1}^n \mathbf{M}_k P_k = [\mathbf{M}_1, \mathbf{M}_2, \dots, \mathbf{M}_{n_a^2}] \begin{bmatrix} P_1 \\ P_2 \\ \vdots \\ P_{n_a^2} \end{bmatrix} \quad (19)$$

with $\mathbf{M}_k \in \mathbb{R}^{n^2 \times n^2}$ being the Toeplitz convolution matrix for the random PSF in the k -th angular view. Light field imaging based on diffuser camera can be implemented with both lens⁸ and lensless manners⁷. When being used with a lens, the PSF for each sub-aperture image is more compactly supported, leading to an efficient utilization of the sensor pixels owing to smaller boarder effects. In contrast, the lensless approach features system simplicity, and it is free from lens-aberrations.

It is now clear that the differentiating factor among existing compressive light field imaging methods is the matrix operating on each sub-aperture image. The matrices (\mathbf{I}, \mathbf{C}_k) in Ashok, Babacan, and Marwah et.al. are all diagonal. As a result, the sensor resolution directly determines the spatial resolution of the recovered light field (both y and P_k are in $\mathbb{R}^{n^2 \times 1}$), making these methods ill-suited for 0D, 1D, and sparse 2D sensors. In contrast, the Toeplitz matrix \mathbf{M}_k in diffuser-camera-based light field imaging is non-diagonal, and its row vectors multiplex multiple elements of P_k into one measurement in y (owing to a nonlocal PSF). Though not being demonstrated yet, this allows in theory the recovery of a 4D light field from a sub-Nyquist measurement dataset (that is $y \in \mathbb{R}^{m \times 1}$ with $m < n^2$ while $P_k \in \mathbb{R}^{n^2 \times 1}$).

In contrast, CLIP is a systematic method for designing and transforming any imaging methods with nonlocal data acquisition into a highly efficient light field imaging approach. For a given imaging model with measurement matrix \mathbf{A} , the transformation of CLIP is achieved by splitting the measurements into different angular views, as illustrated below:

$$y = \mathbf{A}x = \begin{bmatrix} a_1^T \\ a_2^T \\ \vdots \\ \vdots \\ a_l^T \end{bmatrix} x \xrightarrow[\text{CLIP Step 1}]{\text{Transforming: measurement splitting}}$$

$$y = \begin{bmatrix} \text{view}_1 \begin{bmatrix} \mathbf{a}_1^T \\ \vdots \\ \mathbf{a}_q^T \end{bmatrix} & \cdots & \mathbf{0} \\ \vdots & \text{view}_k \begin{bmatrix} \mathbf{a}_{kq+1}^T \\ \vdots \\ \mathbf{a}_{kq+q}^T \end{bmatrix} & \vdots \\ \mathbf{0} & \ddots & \mathbf{0} \\ \mathbf{0} & \cdots & \text{view}_l \begin{bmatrix} \mathbf{a}_{lq+1}^T \\ \vdots \\ \mathbf{a}_{lq+q}^T \end{bmatrix} \end{bmatrix} \begin{bmatrix} P_1 \\ P_2 \\ \vdots \\ P_l \end{bmatrix} = \begin{bmatrix} \mathbf{A}_1 & \cdots & \mathbf{0} \\ \vdots & \mathbf{A}_2 & \vdots \\ \mathbf{0} & \ddots & \mathbf{0} \\ \mathbf{0} & \cdots & \mathbf{A}_l \end{bmatrix} \begin{bmatrix} P_1 \\ P_2 \\ \vdots \\ P_l \end{bmatrix} = \mathbf{A}' \begin{bmatrix} P_1 \\ P_2 \\ \vdots \\ P_l \end{bmatrix}, \quad (20)$$

where \mathbf{a}_k^T is a row vector and x (an image from a single angular view) is extended to a 4D light field (P_1 to P_l) with $l=n_a^2$ views (sub-apertures). While the imaging model becomes block diagonal, recovering the light field is equivalent to solve each sub-aperture image P_k with a corresponding sub-measurement matrix \mathbf{A}_k . We can better exploit the correlations (redundancy) in the 4D light field by solving Supplementary Eq. 20 with appropriate sparsity based regularizations, as used in compressive light field imaging methods³⁻⁵. It is noteworthy that the elemental matrix \mathbf{A}_k is no longer diagonal as \mathbf{I} or \mathbf{C}_k , a key fact that enables CLIP to use 0D or 1D sensors for light field imaging. We demonstrated 4D light field recovery using CLIP in Supplementary Note 7.

The second key differentiating factor of CLIP is explicit modeling of the correlations among sub-aperture images as $P_k = \mathbf{B}_k h$ via light field propagation, assuming a uniform angular intensity distribution as derived in Supplementary Note 1. This simplifies Supplementary Eq. 20 to the CLIP equation 3 in the main text:

$$y = \begin{bmatrix} \mathbf{A}_1 & \cdots & \mathbf{0} \\ \vdots & \mathbf{A}_2 & \vdots \\ \mathbf{0} & \ddots & \mathbf{0} \\ \mathbf{0} & \cdots & \mathbf{A}_l \end{bmatrix} \begin{bmatrix} P_1 \\ P_2 \\ \vdots \\ P_l \end{bmatrix} \xrightarrow[\text{CLIP Step 2}]{P_k = \mathbf{B}_k h} y = \begin{bmatrix} \mathbf{A}_1 & \cdots & \mathbf{0} \\ \vdots & \mathbf{A}_2 & \vdots \\ \mathbf{0} & \ddots & \mathbf{0} \\ \mathbf{0} & \cdots & \mathbf{A}_l \end{bmatrix} \begin{bmatrix} \mathbf{B}_1 h \\ \mathbf{B}_2 h \\ \vdots \\ \mathbf{B}_l h \end{bmatrix} = \begin{bmatrix} \mathbf{A}_1 \mathbf{B}_1 \\ \mathbf{A}_2 \mathbf{B}_2 \\ \vdots \\ \mathbf{A}_l \mathbf{B}_l \end{bmatrix} h = \mathbf{A}'' h. \quad (21)$$

This step has the advantage of enabling more complicated images to be recovered without the need of finding/learning a better sparsifying basis for the 4D light field, which is an important step in Marwah's work. We show this advantage in Supplementary Note 8.

The computation complexity of compressive light field photography and CLIP depends on the light field resolution and the applied regularization method under the framework of regularization by denoising (see **Methods**). In CLIP, each iteration involves a pass of \mathbf{A}'' and \mathbf{A}''^T along with a denoising step. The complexity for the shearing operation and matrix \mathbf{A} is $o(n_a^2 N^2)$ and $o(mN^2)$ respectively, leading to a total complexity of $o((n_a^2 + m)N^2)$ for both \mathbf{A}'' and \mathbf{A}''^T . The complexity of BM3D and TV denoising for regularization is directly related to the image size as $o(kN^2)$, with k being a denoiser-dependent constant. Therefore, the total complexity of CLIP image recovery is $o((2m + 2n_a^2 + k)N^2)$ per iteration. In comparison, while the complexity for \mathbf{A}' and \mathbf{A}'^T in Supplementary Eq. 20 for retrieving the 4D light field remains $o(mN^2)$ owing to the block diagonal structure, the denoising complexity of a 4D light field becomes $o(kn_a^2 N^2)$, resulting in a total complexity of $o((2m + kn_a^2)N^2)$. Similarly, we can analyze the computation complexity per iteration for compressive light field imaging

methods based on the model in Supplementary Eq. 17 to 19. Supplementary Table 1 summarizes the characteristics of CLIP and compressive light field photography. It is worth noting that the computation complexity of Marwah’s work does not account for the dictionary learning process, and the regularization is applied on the entire light field. Also, the convolution model of the diffuser-camera is accelerated by FFT.

Supplementary Table 1 Comparison of CLIP and compressive light field photography

Methods	Sensor	Light field size	Measurement data size	Compression axis	Computation complexity	
Ashok ⁷	2D	$n_a \times n_a \times N \times N$	$r \times N \times N$	Angular or spatial	$o((2r + k)n_a^2 N^2)$	
Babacan ⁸	2D	$n_a \times n_a \times N \times N$	$r \times N \times N$	Angular	$o((2r + k)n_a^2 N^2)$	
Marwah ³	2D	$n_a \times n_a \times N \times N$	$N \times N$	Angular	$o((2 + k)n_a^2 N^2)$	
Cai ⁷ , Antipa ⁸	2D	$n_a \times n_a \times N \times N$	$N \times N$	Angular	$o((4 \log N + kn_a^2)N^2)$	
CLIP	0D, 1D, 2D	$n_a \times n_a \times N \times N$	$m (\leq N \times N)$	Angular and/or spatial	4D light field	$o((2m + kn_a^2)N^2)$
					Refocus image	$o((2m + 2n_a^2 + k)N^2)$

Supplementary Note 8. CLIP 4D light field reconstruction versus direct reconstruction

While CLIP can recover a 4D light field as demonstrated in previous Note, we show here that directly recovering a refocused image can better accommodate complex scenes, particularly for imaging with lower dimension (1D or 0D) sensors. Marwah’s work relied on a dictionary learning process to obtain a representation basis to better sparsify the 4D light field, thereby attaining excellent 4D light field reconstruction for complex scenes. On the other hand, Antipa⁴ pointed out that improper regularization of the 4D light field in diffuser-based camera can degrade (or even destroy) the angular information in the light field.

In contrast, CLIP doesn’t rely on high quality 4D light field reconstruction to obtain excellent refocused images: CLIP’s complementary measurements among sub-apertures can significantly improve the refocused images despite the recovered 4D light field may not be of high quality, which is the case under the compressive regime. Further, CLIP can directly recover a refocused image like coded-aperture and wavefront-coding methods to accommodate complex scenes better, as explained in previous section. We demonstrate this via a synthetic study for the synthetic scene 2 and an experimentally acquired light field from the ‘letter scene’, using a sampling ration of SR=1. During the reconstruction for the 4D light field, the regularization parameter is tuned from to obtain a best refocused image from the light field data. Supplementary Figure 11 shows the recovered 4D light field and refocused images for the two scenes under the CLIP-1D (a and b) and CLIP-0D (c and d) implementations, with the NMSE listed in Supplementary Table 4. It is noted that while the light field suffers from significant background signals and noises, the refocusing processing coherently assembles CLIP’s complementary imaging across the sub-apertures to yield substantially better refocused image. Moreover, CLIP’s direct reconstruction further improved the quality of the refocused image by rendering more image details and a higher contrast.

Supplementary Table 4. NMSE of 4D and direct CLIP reconstructions

	Scene	Method	s=-1.0	s=1.0		Method	s=1.0	s=1.0
CLIP-1D	1	4D recon.	11.98%	6.08%	CLIP-0D	4D recon.	10.47%	10.67%

	Direct recon.	3.66%	4.16%
2	4D recon.	18.68%	8.48 %
	Direct recon.	6.33%	6.75%

	Direct recon.	6.23%	7.35%
	4D recon.	13.02%	7.53%
	Direct recon.	4.41%	4.05%

Supplementary Figure 11. 4D light field reconstruction versus direct reconstruction of refocused images by CLIP. a-b, CLIP-1D reconstruction for the synthetic scene and the experimental ‘letter’ scene. c-d, CLIP-0D reconstruction for the two scenes. The sampling ratio of CLIP is fixed at SR=1.

5. The performance of their method is not evaluated well. In compressed sensing, evaluation of data fidelity is essential. Without comparison of the reconstructed images with the ground truth measured by conventional methods (with a lower acquisition rate), it is impossible to judge if the method is good or not.

Response:

We thank the reviewer for pointing out the importance of quantitatively evaluating the image performance (data fidelity) of CLIP. In the revised manuscript, we quantified the accuracy of CLIP using normalized mean square error (NMSE) with respect to the ground truth in both experiments and synthetic studies. We stressed this point in the

last paragraph of Principle Section of the revised main text that reads “... We quantified the efficacy of CLIP for light field imaging experimentally with a 0D sensor in Supplementary Note 7, and further evaluated the CLIP reconstruction accuracy synthetically with both 0D and 1D sensors in Supplementary Note 11, which employs CLIP to represent custom-acquired 4D light field data for scenes of different complexities and BRDF characteristics...”.

Detailed experimental quantification for CLIP imaging with 0D sensors (CLIP-0D) is given in the dedicated Supplementary Note 7 of the revised Supplementary Materials, and extensive synthetic quantification for both CLIP-0D and CLIP-1D (using a 1D array sensor) are summarized in Supplementary Table 7-10 of Supplementary Note 11 (the original Supplementary Note 6), which uses CLIP to represent the 4D light field data for scenes of different BRDFs. Both Supplementary Note 7 and Supplementary Tables 7-10 are excerpted below.

“ ...

Supplementary Note 7. Quantitative evaluation of CLIP performance in experiments

We quantitatively evaluated the performance of CLIP via experimental measurements when feasible and turned to synthetic studies otherwise. This is because for computational imaging employing nonlocal sampling strategies, ground truth data is typically difficult to obtain experimentally: a system reconfiguration with perfect alignment is necessary. Taking CLIP imaging with 1D sensors for example, one needs to swap the cylindrical lenslet array into its spherical counterpart and adds a 1D scanning to obtain the ground-truth light field. This reference imaging needs to be precisely realigned to show the same magnification and field of view with CLIP: any mismatch will otherwise bias the quantitative evaluation of its imaging accuracy.

For CLIP imaging with 0D sensors, the 4D light field can be fully sampled (though not based on conventional 2D sensors): for each angular position behind the lens, the sub-aperture image can be acquired with a measurement number equal to or larger than the image resolution (thus doesn't rely on compressive sensing), and this imaging process is repeated at all angular positions. CLIP measurement can be readily obtained from this dataset by extracting a small subset measurement from each angular position and stacking the complementarily extracted data into a final measurement as described by Supplementary Eq. 20. We present experimental validation of CLIP with 0D sensor in this section and synthetic evaluation of CLIP with 1D sensor in the following sections.

Two different scenes composed of printed letters were imaged by CLIP-0D experimentally, and both the 4D light field and direct image reconstructions are demonstrated under different sampling ratio SR. The ground truth 4D light field has a resolution of $4 \times 4 \times 128 \times 128$ and was obtained by reconstruct each sub-aperture image using a complete measurement. Similarly, the ground truth refocused image was obtained from the 4D light field. Supplementary Figure 8 and 9 shows the 4D light field reconstruction results by CLIP for the two scenes, and the direct reconstruction of different refocused images are given in Supplementary Figure 10. The reconstruction error is quantified by NMSE in Supplementary Table 3. It is noted that both the 4D light field and direct reconstruction of refocused images attained a NMSE error below 10% in experiments.

Supplementary Table 3. NMSE of CLIP-0D reconstruction for experimental scenes

	Sampling ratio	4D light field	$s = -1$	$s = -0.3$	$s = 0.3$	$s = 1$
Scene 1	$SR = 1$	7.08%	1.94%	1.53%	1.39%	1.76%
	$SR = 0.5$	7.31%	2.32%	2.15%	1.83%	2.29%
	$SR = 0.25$	8.84%	5.41%	3.06%	2.57%	3.22%
Scene 2	$SR = 1$	1.34%	0.94%	0.66%	0.61%	0.67%
	$SR = 0.5$	2.13%	1.34%	1.39%	1.35%	1.46%
	$SR = 0.25$	5.06%	3.99%	4.16%	3.75%	3.07%

Supplementary Figure 8. CLIP-0D 4D light field reconstruction for experimental scene 1. The sampling ratio of CLIP is varied from $SR = 1$ to 0.25.

Supplementary Figure 9. CLIP-0D 4D light field reconstruction for experimental scene 2. The sampling ratio of CLIP is varied from $SR = 1$ to 0.25.

Supplementary Figure 10. CLIP-0D direct reconstruction of refocused images for experimental scene 1 and 2. The sampling ratio of CLIP is varied from SR= 1 to 0.25.

...”

“ ...

Supplementary Note 11. CLIP generality: representing 4D light field data

Supplementary Table 7. NMSE of CLIP representation of the ‘letters’ scene

	SR	2	1	0.5		SR	2	1	0.5	0.25
CLIP -1D	s= -1.0	1.75%	2.79%	5.32%	CLIP -0D	s= -1.0	1.75%	2.60%	3.99%	6.61%
	s= -0.4	1.47%	2.63%	4.54%		s= -0.4	1.48%	2.49%	4.06%	5.58%
	s= -0.1	1.59%	3.04%	5.91%		s= -0.1	1.64 %	2.84%	4.48%	7.50%

Supplementary Table 8. NMSE of CLIP representation of the ‘toy-bottle’ scene

	SR	2	1	0.5		SR	2	1	0.5	0.25
CLIP -1D	s= -0.3	0.87%	1.65%	2.37 %	CLIP -0D	s= -0.3	1.95%	3.70%	5.00%	7.86%
	s= 0.0	1.21%	2.40%	3.19%		s= 0.0	1.76%	4.99%	5.70%	6.83%
	s= 0.3	1.42%	2.86%	3.65%		s= 0.3	2.56%	4.01%	6.24%	9.38%

Supplementary Table 9. NMSE of CLIP representation of the ‘slanted-text’ scene

	SR	2	1	0.5		SR	2	1	0.5	0.25
CLIP -1D	s= -1.0	0.97%	4.5%	9.97 %	CLIP -0D	s= -1.0	3.01%	4.91%	8.03%	11.94%
	s= -0.4	0.80 %	1.78%	6.41%		s= -0.4	4.10%	3.85%	6.13%	9.31%

s= 0.3	0.72%	3.39%	5.54%	s= 0.3	3.29%	5.88%	7.62%	15.77%
--------	-------	-------	-------	--------	-------	-------	-------	--------

Supplementary Table 10. NMSE of CLIP representation of the ‘bolt-letter’ scene

	SR	2	1	0.5		SR	2	1	0.5	0.25
CLIP -1D	s= -1.0	0.97%	2.46%	4.77 %	CLIP -0D	s= -1.0	0.82%	1.27%	1.96%	3.28%
	s= -0.5	0.80 %	2.04%	3.84%		s= -0.5	1.31%	1.72%	2.35%	3.25%
	s= 0.2	0.72%	1.70%	3.23%		s= 0.2	0.85%	1.22%	2.11%	2.94%

...”

6. The authors' main claim "enable snapshot 3D imaging with an extended depth range and through severe scene occlusions" in the abstract is suspicious. In the supplementary movies, the shape of the objects significantly changes when they are occluded. Again, quantitative evaluation image reconstruction is needed for supporting their claim.

Response:

We improved the experiments of imaging through occlusions. In previous demonstration, the CLIP measurement across the lenslet array was not sufficiently randomized: the projection angle of the cylindrical lenslet array were uniformly spaced along the array direction. Under occlusions, the object only yields a subset of the full measurement entries. To maximize (in a statistical sense) the incoherence among the measurement subset at any time instant (the occlusion changes dynamically with object motion), it is best to randomly distribute the projection angles of the cylindrical lens along the array direction. This is similar in spirit to use a random subset of Fourier basis for compressive single pixel or MRI imaging: the Fourier basis needs be randomly shuffled first and then chose an arbitrary subset from it.

Guided by this principle, we further randomized the cylindrical lens angles along the array direction in the new experiments, and improved the imaging quality for imaging through occlusions. Regarding this, we revised the Methods section to stress the randomized cylindrical lenslet arrangement that reads “ ... **For optimal imaging through occlusions, the cylindrical lenslet angles are further randomly distributed along the lenslet array direction, as the effective measurement entries for the occluded objects are reduced to a subset of the measurement entry in the imaging model. Such random distribution maximizes statistically the incoherence among any subset of the measurements to ensure consistent image recovery performance...**”.

The improved imaging results (along with the corresponding Supplementary Video 1) are revised in Fig. 2d in the main text, which is appended below.

Fig. 2. 3D imaging through occlusions. d Three representative frames of imaging a 2×2 grid pattern moving across the CLIP camera FOV behind a rectangular occluder. Note that signals from the black occluders are enhanced relative to the objects for better visualization.

Furthermore, we also revised Supplementary Note 10 of Supplementary Materials to quantitatively evaluate the performance of CLIP in imaging through occlusions via synthetic studies, which shows that CLIP can achieve a small error (mostly $<10\%$ NMSE) for seeing through occlusions with orders of magnitude less data. We clarified this point in the last sentence of '3D imaging through Occlusions' section in the revised main text that reads "... We further quantified the accuracy of CLIP imaging through occlusions via synthetic studies in Supplementary Note 10, which shows a small imaging error (\$<10\%\$ ) can be obtained by CLIP despite of a large reduction (\$>100\$ times) in light field measurement data...".

We excerpted Supplementary Note 10 below for a detailed clarification.

"... We further compare CLIP with conventional light field imaging for seeing through occlusions via synthetic studies. The 4D light field for 3D scenes were rendered in Blender software with a resolution of $8 \times 8 \times 128 \times 128$, and CLIP measurement were obtained as in previously sections. Unlike ToF based measurements that can separate signals of the occluder and occluded objects in time, conventional imaging systems can only defocus the occluder, yielding significant background for visualizing the occluded objects. To emulate ToF measurement for minimizing background, the occluder can be made black in Blender such that its image signal is negligible in the generated light field. Supplementary Figure 14 shows four examples of imaging through occlusions: a mannequin standing behind a tree, the mannequin partially occluded by the black rectangular plate, the 'CLIP' letter placed behind a bush, and the 'CLIP' letter being blocked by a black rectangular occluder. The CLIP reconstruction NMSE errors are shown in Supplementary Table 6. It is noted that even with a sampling ratio of $SR=0.5$ that corresponds to a reduction of the 4D light field by 128 times, CLIP can effectively see through severe occlusions with an error below 10%. With ToF measurement that produces far sparser 2D instantaneous images and separates the occluder signal in time, as

emulated by black occluder, CLIP can hence attain background-free imaging of occluded objects with a small number of sensors.

Supplementary Table 6. NMSE of CLIP imaging through occlusions

	Scene	SR=1.0	SR=0.5		Scene	s=1.0	s=1.0
CLIP-1D	1	4.07%	6.80%	CLIP-0D	1	3.59%	5.20%
	2	4.99%	9.12%		2	10.17%	14.0%
	3	3.03%	4.83%		3	4.37%	6.41%
	4	2.62%	4.12%		3	3.25%	4.77%

[redacted]

Supplementary Figure 14. CLIP imaging through occlusions for four different scenes. The sampling ratio of CLIP is varied from SR= 1 to 0.5.

...”

Referencess

1. Levoy, M. Light Fields and Computational Imaging. *Computer* **39**, 46–55 (2006).
2. Ng, R. DIGITAL LIGHT FIELD PHOTOGRAPHY. 203.

3. Marwah, K., Wetzstein, G., Bando, Y. & Raskar, R. Compressive light field photography using overcomplete dictionaries and optimized projections. *ACM Trans. Graph.* **32**, 46:1-46:12 (2013).
4. Antipa, N., Necula, S., Ng, R. & Waller, L. Single-shot diffuser-encoded light field imaging. in *2016 IEEE International Conference on Computational Photography (ICCP)* 1–11 (2016). doi:10.1109/ICCPHOT.2016.7492880.
5. Cai, Z. *et al.* Lensless light-field imaging through diffuser encoding. *Light Sci Appl* **9**, 143 (2020).
6. Antipa, N. *et al.* DiffuserCam: lensless single-exposure 3D imaging. *Optica, OPTICA* **5**, 1–9 (2018).
7. Ashok, A. & Neifeld, M. A. Compressive light field imaging. in *Three-Dimensional Imaging, Visualization, and Display 2010 and Display Technologies and Applications for Defense, Security, and Avionics IV* vol. 7690 221–232 (SPIE, 2010).
8. Babacan, S. D. *et al.* Compressive Light Field Sensing. *IEEE Transactions on Image Processing* **21**, 4746–4757 (2012).
9. Liu, X., Bauer, S. & Velten, A. Phasor field diffraction based reconstruction for fast non-line-of-sight imaging systems. *Nature Communications* **11**, 1–13 (2020).
10. Manna, M. L., Nam, J.-H., Reza, S. A., Velten, A. & Velten, A. Non-line-of-sight-imaging using dynamic relay surfaces. *Opt. Express, OE* **28**, 5331–5339 (2020).
11. Lindell, D. B., Wetzstein, G. & O'Toole, M. Wave-based Non-line-of-sight Imaging Using Fast F-k Migration. *ACM Trans. Graph.* **38**, 116:1-116:13 (2019).
12. Bastounis, A. & Hansen, A. C. On the absence of the RIP in real-world applications of compressed sensing and the RIP in levels. *arXiv:1411.4449 [cs, math]* (2015).
13. Roman, B., Bastounis, A., Adcock, B. & Hansen, A. C. On fundamentals of models and sampling in compressed sensing. 12.
14. Baraniuk, R. G., Cevher, V., Duarte, M. F. & Hegde, C. Model-Based Compressive Sensing. *IEEE Trans. Inform. Theory* **56**, 1982–2001 (2010).

REVIEWERS' COMMENTS

Reviewer #1 (Remarks to the Author):

I share the concerns of reviewer 3 that, while containing some interesting approaches, this is not dramatically different from existing compressive light field ideas. The added analysis definitely improves the clarity and content of the paper.

The noise model used seems to be additive gaussian noise. With modern cameras and under normal light levels and exposure times, measurements are likely poisson noise limited. Poisson noise is not additive and often results in very different behavior compared to additive gaussian noise, especially in coding or compressed sensing systems. It would be good to model poisson noise where noise is modeled in the simulations.

Overall, I am not sure whether the scope of the contribution warrants publication in Nature Communications, with the substantial added materials, I think it can be publishable. There is nothing technically wrong with the paper.

Reviewer #2 (Remarks to the Author):

The authors have significantly revised their manuscript. I think it is suitable for publication in Nature Communications.

Reviewer #3 (Remarks to the Author):

The authors have addressed my comments reasonably well. I recommend acceptance of this work for publication.

Response Letter

Reviewer 1

I share the concerns of reviewer 3 that, while containing some interesting approaches, this is not dramatically different from existing compressive light field ideas. The added analysis definitely improves the clarity and content of the paper.

The noise model used seems to be additive gaussian noise. With modern cameras and under normal light levels and exposure times, measurements are likely poisson noise limited. Poisson noise is not additive and often results in very different behavior compared to additive gaussian noise, especially in coding or compressed sensing systems. It would be good to model poisson noise where noise is modeled in the simulations.

Overall, I am not sure whether the scope of the contribution warrants publication in Nature Communications, with the substantial added materials, I think it can be publishable. There is nothing technically wrong with the paper.

Response:

We thank the reviewer for the constructive comments on modelling Poisson noises in simulation studies, which are added in revised Supplementary Note 9 (Figure 13 and Table 6, excerpted below for reference), showing our method works well under Poisson-noise-limited applications.

Regarding the difference with existing compressive light field cameras, we would like to highlight again our work's is not necessarily compressive, and it is a unique framework that enables sensors of arbitrary formats for efficient recording of 4D light field. We revised the sentence in Discussion section as "The CLIP framework encompasses and goes far beyond these methods to utilize sensors of arbitrary formats for efficient light field imaging (**compressive or not**), with a flexible nonlocal sampling strategy that promotes imaging robustness and better exploitation of the sparsity characteristic of high-dimensional data."

Revisions in Supplementary Note 9 is appended below in red:

The robustness of CLIP for photon-starved imaging applications, which are limited by Poisson (or shot) noises, are demonstrated in Supplementary Figure 13 by varying the maximum number of photons in measurement from 400 to 10000. As indicated by the NMSE in Supplementary Table 6, while CLIP-0D is more susceptible to Poisson noises, it can still recover the rough structure of complex scenes with a maximum of only 1000 photons. Since single pixel imaging usually benefit from a larger photon-detector, CLIP-0D is expected to cope well with shot-noise limited imaging applications.

Supplementary Table 6. NMSE of CLIP imaging with different number photons

	Photons	Scene 1	Scene 2		Photons	Scene 1	Scene 2
CLIP-1D	400	7.30%	2.85%	CLIP-0D	400	57.88%	43.45%
	1000	6.54%	2.66%		1000	40.62%	25.45%
	4000	5.92%	2.55%		4000	20.58%	16.16%
	10000	6.01%	2.53%		10000	14.95%	7.05%

Supplementary Figure 13. CLIP reconstruction with different number of photons. NP is the maximum number of photons in the measurement dataset.

Reviewer 2

The authors have significantly revised their manuscript. I think it is suitable for publication in Nature Communications.

Response:

We appreciate the reviewer's positive comments on our work.

Reviewer 3

The authors have addressed my comments reasonably well. I recommend acceptance of this work for publication.

Response:

We thank the reviewer for the positive comments on our work.